# Dense SAE Latents Are Features, Not Bugs

**Xiaoqing Sun**[*]
MIT

**Alessandro Stolfo**[*]
ETH Zürich

**Joshua Engels**
MIT

**Ben Wu**
University of Sheffield

**Senthooran Rajamanoharan**

**Mrinmaya Sachan**
ETH Zürich

**Max Tegmark**
MIT

## Abstract

Sparse autoencoders (SAEs) are designed to extract interpretable features from language models by enforcing a sparsity constraint. Ideally, training an SAE would yield latents that are both sparse and semantically meaningful. However, many SAE latents activate frequently (i.e., are *dense*), raising concerns that they may be undesirable artifacts of the training procedure. In this work, we systematically investigate the geometry, function, and origin of dense latents and show that they are not only persistent but often reflect meaningful model representations. We first demonstrate that dense latents tend to form antipodal pairs that reconstruct specific directions in the residual stream, and that ablating their subspace suppresses the emergence of new dense features in retrained SAEs—suggesting that high density features are an intrinsic property of the residual space. We then introduce a taxonomy of dense latents, identifying classes tied to position tracking, context binding, entropy regulation, letter-specific output signals, part-of-speech, and principal component reconstruction. Finally, we analyze how these features evolve across layers, revealing a shift from structural features in early layers, to semantic features in mid layers, and finally to output-oriented signals in the last layers of the model. Our findings indicate that dense latents serve functional roles in language model computation and should not be dismissed as training noise.

## 1 Introduction

Sparse autoencoders (SAEs) are an unsupervised method for extracting interpretable features from language models [Bricken et al., 2023, Huben et al., 2024, Kissane et al., 2024]. They address the challenge of polysemanticity, where individual neurons activate in semantically diverse contexts that defy a single explanation [Olah et al., 2017, Elhage et al., 2022]. SAEs are trained to reconstruct the activations of a language model under a sparsity constraint applied to a bottleneck layer, ensuring that only a small subset of latents are active at a time.[2] This method effectively recovers interpretable features in a variety of models, including Claude 3 Sonnet [Templeton et al., 2024] and GPT-4 [Gao et al., 2025].

Ideally, a trained SAE would yield a large set of interpretable and sparsely activating latents. In practice, however, SAEs exhibit a substantial fraction of densely activating latents, activating on 10% to 50% of tokens [Cunningham and Conerly, 2024, Rajamanoharan et al., 2024b]. These dense latents

---

[*]Equal contribution. Correspondence to `xqsun@mit.edu` and `stolfoa@ethz.ch`.
[2]We use "latent" to refer to an entry in the SAE's sparse hidden layer.

39th Conference on Neural Information Processing Systems (NeurIPS 2025).

are challenging to interpret based solely on their activation patterns. It remains unclear whether they arise as an optimization by-product, or if they instead capture inherently dense signals present in the model's residual stream [Chen and Batson, 2025, Rajamanoharan et al., 2025].

In this work, we investigate several properties of dense SAE latents and the residual stream subspaces they span, uncovering evidence that these latents track meaningful residual stream information. First, we observe that when retraining an SAE on model activations with the dense latent space ablated, virtually no dense latents are learned—dense latents reflect an intrinsic property of the residual stream rather than a training artifact. We then study the geometry of dense latents and observe that they tend to form antipodal pairs, with each pair effectively reconstructing a single direction.

We then examine the Gemma Scope suite of SAEs [Lieberum et al., 2024] across layers to propose a taxonomy of dense latents. We identify latents whose activations encode positional information, latents reconstructing a subspace of the residual stream linked to entropy regulation [Stolfo et al., 2024, Cancedda, 2024], latents tracking high-level shifts in the text, latents encoding letter-specific output signals, latents tracking parts of speech, and latents reconstructing the first residual stream principal component direction. We additionally examine how these dense latents transform across layers, finding that there is a pronounced increase in the number of dense latents just before the unembedding, as well as a shift from structural signals in early layers (e.g., position tracking) to output-oriented signals at the end. Our findings provide evidence that dense SAE latents reflect inherently dense mechanistic functions within language models.

## 2   Background

**SAEs.**   Sparse autoencoders (SAEs) are trained to reconstruct a language model's activations $\mathbf{x} \in \mathbb{R}^{d_{\text{model}}}$ while imposing a sparsity constraint [Yun et al., 2021, Huben et al., 2024]. This computation can be represented as:

$$\mathbf{f}(\mathbf{x}) := \sigma(\mathbf{W}_{\text{enc}}\mathbf{x} + \mathbf{b}_{\text{enc}}),$$
$$\hat{\mathbf{x}}(\mathbf{f}) := \mathbf{W}_{\text{dec}}\mathbf{f} + \mathbf{b}_{\text{dec}},$$

where $\mathbf{f}(\mathbf{x}) \in \mathbb{R}^{d_{\text{sae}}}$ is a sparse, non-negative vector of latents, with $d_{\text{sae}} \gg d_{\text{model}}$, and $\sigma$ is a non-linear activation function. SAEs are typically trained to minimize the L2 distance between the original activation and its reconstruction $\|\mathbf{x} - \hat{\mathbf{x}}(\mathbf{f}(\mathbf{x}))\|_2^2$ while a sparsity constraint is imposed on $\mathbf{f}$ by adding a sparsity-related loss component or via specific activation functions. We denote the encoder and decoder weights of the latent at index $i$ as $\mathbf{W}_{\text{enc}}^{(i)}$ and $\mathbf{W}_{\text{dec}}^{(i)}$, respectively. Unless noted otherwise, we use "dense" to refer to latents with an activation frequency larger than 0.1.

**Experimental Setup.**   We focus our investigation on the Gemma Scope SAEs [Lieberum et al., 2024] trained on Gemma 2 2B [Gemma Team, 2024], which use a JumpReLU activation function [Rajamanoharan et al., 2024b]. We additionally train TopK SAEs [Gao et al., 2025] on 1B tokens of the OpenWebText corpus [Gokaslan and Cohen, 2019] for our experiments in §3.1.[3] Activation densities for Gemma Scope latents are from Neuronpedia [Lin, 2023], while densities for our TopK SAEs are computed over 100M tokens from the C4 Corpus [Raffel et al., 2020]. Full experimental details are in Appendix B.

## 3   General Properties of Dense Latents

We begin by examining structural properties of dense SAE latents, finding that they arise from a specific residual stream subspace (§3.1), and that they tend to cluster in antipodal pairs (§3.2).

### 3.1   Dense Latents Reflect Intrinsic Properties of the Residual Stream

To determine whether dense SAE latents arise from the training procedure or reflect an intrinsic property of the residual-stream subspace they reconstruct, we perform a targeted ablation experiment. We identify the subspace spanned by the dense latents of an SAE trained on layer 25 of Gemma 2 2B, then train a new SAE on activations in which this subspace has been zero-ablated. For comparison,

---

[3]We choose TopK for its reliable training and competitive reconstruction–sparsity trade-off.

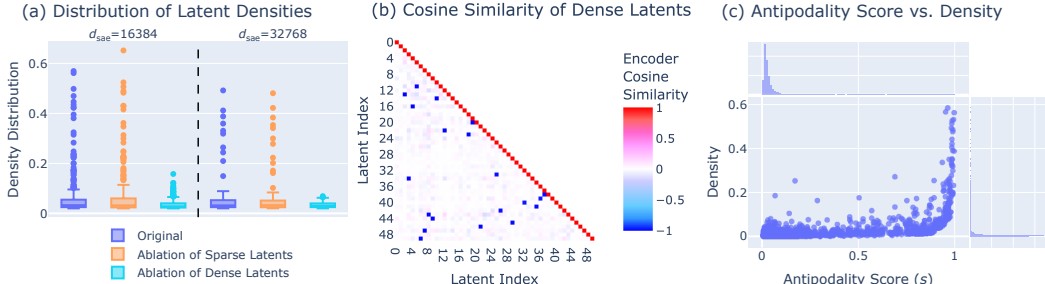

Figure 1: **General Properties of Dense SAE Latents.** (a) Ablating the dense-latent subspace (teal) reduces high-density latents compared to the original (blue) and sparse-latent ablations (orange). (b) Encoder cosine similarity between the top 50 latents with highest density. (c) Dense latents exhibit high antipodality score: they form pairs that reconstruct specific residual stream directions.

we also select an equally sized set of non-dense latents and train a third SAE after ablating their subspace. We repeat this for two dictionary sizes ($d_{\text{sae}} = 16384$ and $32768$).

Figure 1a shows the resulting distributions of latent activation densities. In both dictionary sizes, ablating the dense-latent subspace (teal) yields much fewer high-density latents than the original SAE (blue) and the non-dense ablation (orange). This result implies that densely activating latents are not mere training artifacts but instead track a dense residual-stream subspace whose presence drives the emergence of dense latents. As additional evidence that dense latents are not training artifacts, in Appendix A.2 we show that longer training does not reduce the number of dense latents. We further replicate this dense-subspace ablation on GPT-2 and LLaMA 3.2 with the same outcome (Appendix A.6).

## 3.2 Dense Latents Cluster in Antipodal Pairs

We now examine the geometry of dense latents and observe that they tend to form antipodal pairs. That is, as shown in Figure 1b, there exist many pairs of two dense latents that have nearly opposite decoder vectors (we find a similar result for encoder vectors). This suggests that the SAE allocates two latents in the dictionary to represent a 1-dimensional line.

To quantify whether this phenomenon is specific to dense latents, we introduce an antipodality score $s_i$ for a latent $i$. We first compute the pairwise cosine similarities between the latent's weights (both encoder and decoder) and those of all other latents. Then, we compute the maximum product of encoder and decoder cosine similarity across all pairs $(i, j)$ for all $i \neq j$. Formally, we have

$$s_i := \max_{j \neq i} \left( \text{sim}\left(\mathbf{W}_{\text{enc}}^{(i)}, \mathbf{W}_{\text{enc}}^{(j)}\right) \cdot \text{sim}\left(\mathbf{W}_{\text{dec}}^{(i)}, \mathbf{W}_{\text{dec}}^{(j)}\right) \right), \tag{1}$$

where $\text{sim}(u, v)$ denotes the cosine similarity between vectors $u$ and $v$. This score reflects the extent to which latent $i$ forms an antipodal pairing with another latent: high values of $s_i$ indicate that there is another latent $j$ with both encoder and decoder weights nearly opposite in direction to those of $i$.[4]

As shown in Figure 1c, $s_i$ and the activation density of latent $i$ are strongly positively correlated. The majority of dense latents—particularly those with an activation frequency exceeding 0.3—exhibit pairwise scores greater than 0.9, supporting our conclusions above. We

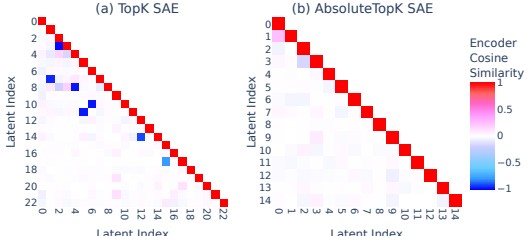

Figure 2: **AbsoluteTopK SAEs show no antipodality.** Allowing the SAE to have both positive and negative latent activations removes antipodal dense latents.

provide density-antipodality visualizations for additional SAEs in Appendix A.1, showing that this trend holds consistently across SAE architectures (JumpReLU and TopK), models (GPT-2 and Gemma), and layers.

---

[4]Although high values of $s$ could be produced by two nearly identical latents, retaining such a pair would be redundant–a scenario we do not observe. Evidence for this is provided in Appendix A.4.

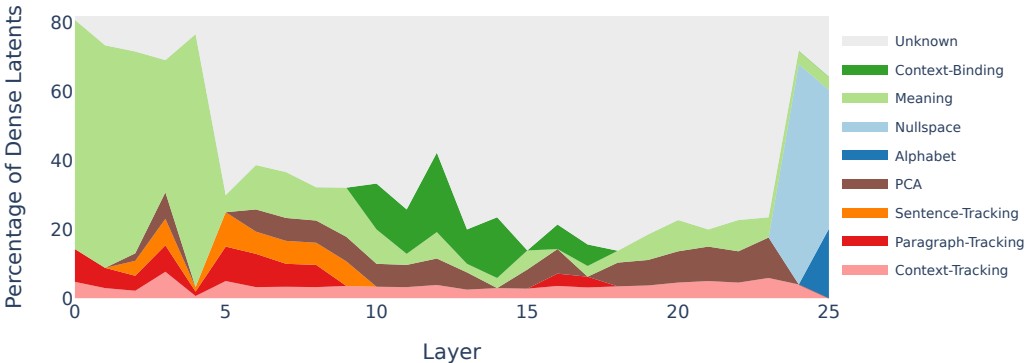

Figure 3: **An overview of our taxonomy of dense latents, for every layer.** See Appendix E.1 for how we created this plot.

Additionally, we train an AbsoluteTopK SAE, which allows activations of SAE latents to be negative, and enforces sparsity by taking the TopK latents with greatest absolute activations. This effectively allows the same latent direction to be used in both "positive" and "negative" directions for reconstruction. We compare this to a TopK SAE trained with the same seed, and show that this AbsoluteTopK activation function eliminates the antipodal dense latents (Figure 2).

## 4 Taxonomy

Having established that dense latents are persistent and geometrically structured, we now investigate their interpretability. We identify classes of dense latents based on the model signals they represent:

- **Position latents** (§4.1) fire based on token position relative to structural boundaries (start of sentence, paragraph or context) and appear early in the network.
- **Context-binding latents** (§4.2) represent context-dependent semantic content and exhibit coherent chunk-level activations, potentially representing high-level ideas within the context.
- **Nullspace latents** (§4.3) track components of the residual stream that have minimal impact on next token prediction. They instead regulate prediction entropy.
- **Alphabet latents** (§4.4) promote broad sets of tokens sharing an initial character.
- **Meaningful-word latents** (§4.5) have activations related to the token part-of-speech tag.
- **PCA latents** (§4.6) lie almost completely within the first PCA components of the activation space.

### 4.1 Position Latents

We first identify a class of dense latents whose activations track the current token's position relative to specific text boundaries. **Context-tracking** latents track token position w.r.t. the BOS token, **paragraph-tracking** latents track token position w.r.t. a paragraph start, and **sentence-tracking** latents track token position w.r.t. a sentence beginning. Context-position latents are similar to "position neurons" from prior work [Gurnee et al., 2024]; the other categories are to the best of our knowledge novel.

To find these latents systematically, we use Spearman's rank correlation coefficient $\rho$. For each dense latent, we capture the projections[5] of the residual stream activations onto its decoder vector for 5000 1024-token-long contexts. We find $\rho$ between this projection and the distance from the last period, the last newline, and the beginning of the input. These boundaries act as proxies for "beginning of sentence", "beginning of paragraph", and "beginning of context", respectively.

---

[5]We use the projection of the residual stream rather than the JumpReLU activations of these latents since we hypothesize that the *direction* itself encodes the positional information, regardless of whether the magnitude exceeds the learned JumpReLU threshold.

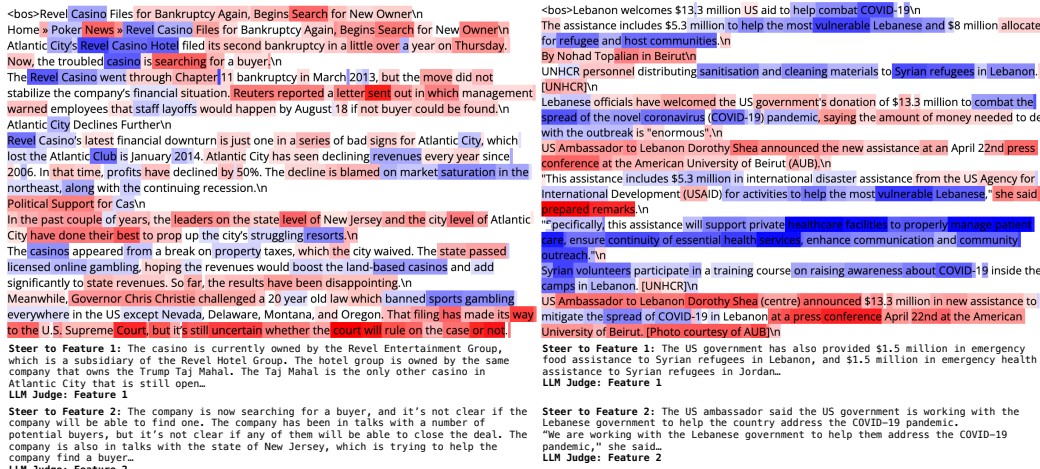

Figure 4: **Context-Binding Latents.** Activation patterns of layer 12 antipodal pair 7541 (blue, feature 1) and 2009 (red, feature 2). In the first context, they seem to be tracking "casino facts" vs "looking for a buyer", while in the second context, they seem to be tracking "healthcare" vs "press conference". Their corresponding completions are in line with the concepts they activated on.

Figure 3 shows the resulting trends: sentence-tracking and paragraph-tracking latents are prominent before layer 10, while context-position tracking latents are present throughout the model. Figure 15 shows $\rho$ for all latents across layers. We can clearly see groups of outlier latents for each category, and thus classify latents as belonging to that category if $|\rho| > 0.4$. Indeed, examples in Appendix E.2 confirm that the identified outlier latents have position-tracking behavior. Notably, Appendix E.2 also shows that paragraph-tracking latents are agnostic to artificially adding formatting newlines, suggesting that this direction in the model tracks true semantic paragraph breaks. Thus, our "distance to newline objective" is just a proxy. We also note that latents with high $\rho$ with periods also have high $\rho$ with newlines, since newlines and periods are correlated in text. In Figure 17, we thus show the $\rho$ for sentence-tracking vs. paragraph-tracking across all dense latents.

At a higher level, it makes sense that the model represents these features in a dense way: positional information is always relevant to the model's predictions (e.g., it must track how far it is in a sentence to correctly predict a period), so the model might store this representation in a consistent direction in every hidden state, which is then learned by the SAE.

## 4.2 Context-Binding Latents

We next identify a class of dense latents that encode different semantic concepts depending on context. Unlike interpretable sparse SAE latents typically associated with fixed meanings, such as the "Golden Gate Bridge" feature in Claude [Templeton et al., 2024], these dense latents appear to *bind* to the main ideas of the context.

We first observe that some dense latents, particularly in middle layers, activate on long consecutive "chunks" of tokens.[6] Examining the activations of such latents, we notice empirically that such latents fire on highly specific concepts *within a context*, but the concepts *vary across contexts*. We generate explanations of these latents with an LLM and confirm that they seem to be more context-specific than sparse latents (see Appendix E.3).

One possible interpretation is that these latents represent general but abstract, difficult-to-interpret properties. However, we also observe that within an antipodal pair, the active latent often switches when the main topic or entity in the text changes (Figure 4, Appendix E.4). This raises the hypothesis that such directions act as "registers" in the residual stream for tracking the active concept, rather than simply representing generic properties.

---

[6]While positional latents also exhibit consecutive activations, here we refer to non-positional latents whose activations cannot be explained by position alone.

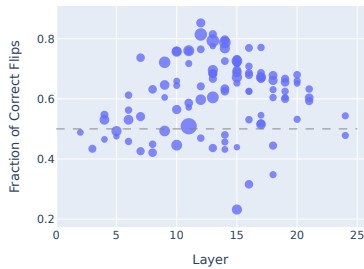

Figure 5: **Fraction of correct flips when steering**, for all latent pairs that have at least one latent $f > 0.2$, and $\geq 40$ flips. Points are sized by number of flips.

| Layer | Latent Pair | In-context | Out-of-context |
|---|---|---|---|
| 12 | (14906, 14599) | 0.051 | 0.717 |
| 12 | (2291, 13295) | 0.028 | 0.760 |
| 12 | (7541, 2009) | 0.043 | 0.711 |
| 13 | (3517, 46) | 0.036 | 0.742 |
| 13 | (15275, 11449) | 0.029 | 0.704 |
| 13 | (12613, 7655) | 0.028 | 0.531 |
| 14 | (11575, 2411) | 0.047 | 0.798 |
| 14 | (8515, 15297) | 0.041 | 0.603 |
| 14 | (6699, 1802) | 0.037 | 0.678 |
| 16 | (2889, 8811) | 0.024 | 0.665 |
| 17 | (10495, 491) | 0.051 | 0.669 |

Table 1: **Fraction of "unclear" judgments** using in-context examples versus out-of-context examples, for the highest-scoring latents by flips.

We thus perform a steering experiment to find the causal effect of these directions. For each antipodal pair (F1, F2), we prompt Gemma 2 2B with input text from the RedPajama dataset [Weber et al., 2024] and generate completions without steering, steering to F1, and steering to F2. An LLM judge [Gemini Team, 2025] is then asked whether each completion is more in line with activating examples (from the input context) of F1 or F2, or unclear. Further details of the methodology are in Appendix E.5.

Since the unsteered generation may already favor F1 or F2, we quantify steering success by the fraction of *flips* from the unsteered judgment that align correctly with the steering direction. For several mid-layer latent pairs, steering reliably shifts completions towards the specific concept previously associated with the latent *in that context* (Figure 5). However, when judged against out-of-context examples, the rate of unclear judgments rises sharply (Table 1). While difficult to rule out the possibility that these directions encode "general uninterpretable" features, the specificity of the steered generation in bringing up context-related ideas suggests that these latents could bind to concepts in a context-dependent, rather than globally consistent, way.

Previous works have uncovered "binding mechanisms" that help the model keep track of in-context associations between entities [Feng and Steinhardt, 2024, Feng et al., 2024]. While our findings do not directly prove such a mechanism, they raise the possibility that dense subspaces may play a similar functional role, distinguishing the currently active semantic concept. Further work could explore the circuits [Marks et al., 2025] involving such subspaces, and challenge the assumption of globally monosemantic directions.

### 4.3 Nullspace Latents

Previous work has identified a $\mathbf{W}_{\mathrm{U}}$ *quasi-nullspace*–the subspace spanned by the last singular vectors of the unembedding matrix $\mathbf{W}_{\mathrm{U}}$–which accounts for a substantial portion of the residual stream's norm, yet has little direct impact on next-token prediction [Cancedda, 2024]. Since this subspace carries high norm, we hypothesize that some dense SAE latents are allocated specifically to reconstruct it.

To test this, we compute the singular value decomposition $\mathbf{W}_{\mathrm{U}} = \mathbf{U}\boldsymbol{\Sigma}\mathbf{V}^{\mathrm{T}}$. Then, we study the composition of an SAE latent $i$'s encoder weight with the space spanned by the last $k$ left singular vectors $\mathbf{U}_{-k}, \ldots, \mathbf{U}_{-1}$ of $\mathbf{W}_{\mathrm{U}}$ by computing the fraction $\rho_k$ of the norm of its encoder weight $\mathbf{W}_{\mathrm{enc}}^{(i)}$ that lies in this subspace:

$$\alpha_k = \frac{\sum_{j=1}^{k} \mathbf{U}_{-j}^{\mathrm{T}} \mathbf{W}_{\mathrm{enc}}^{(i)}}{\|\mathbf{W}_{\mathrm{enc}}^{(i)}\|}. \tag{2}$$

A histogram of $\alpha_{10}$ for the SAE trained at layer 25 of Gemma 2 2B (Figure 6a) shows that 99.6% of latents have $\alpha_{10} < 0.2$. We designate those with $\alpha_{10} > 0.2$ as *nullspace-aligned*. Interestingly, 75% of them are high-density, and account for 40% of the high-density latents in the SAE.

Unlike other dense latents, nullspace-aligned latents are hard to interpret via their token-level activation patterns. Additionally, the tokens they promote are typically uninterpretable "under-trained"

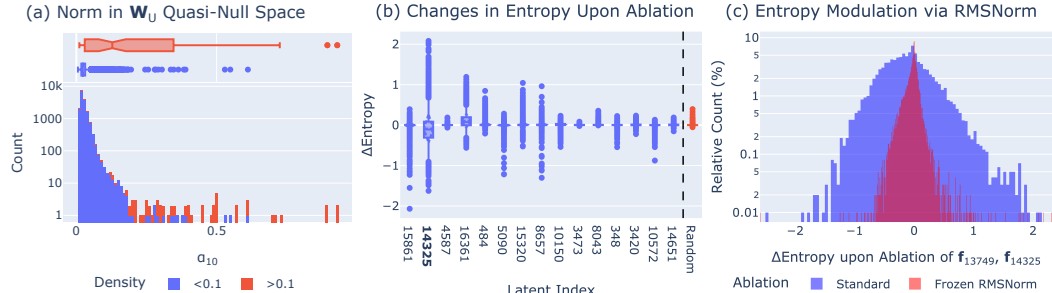

(a) Norm in $\mathbf{W}_U$ Quasi-Null Space      (b) Changes in Entropy Upon Ablation      (c) Entropy Modulation via RMSNorm

Figure 6: **Nullspace Latents.** (a) A small fraction of latents concentrate norm in the final 10 singular directions of $\mathbf{W}_U$, with high-density latents overrepresented in this group. (b) A pair of such latents correlates strongly with model output entropy. (c) Ablating this pair lowers entropy; the effect substantially decreases when RMSNorm scaling is frozen.

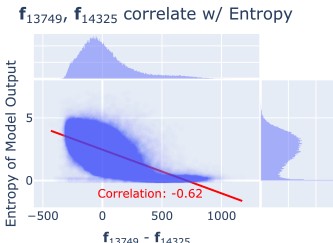

$\mathbf{f}_{13749}$, $\mathbf{f}_{14325}$ correlate w/ Entropy

Figure 7: **Entropy Correlation.** A pair $\mathbf{W}_U$ nullspace-aligned correlates strongly with model output entropy.

| Index | Letter | Density | Metric | Top Tokens |
|-------|--------|---------|--------|-----------|
| 15287 | R | 0.16 | 0.98 | _RI, _rb, getR, _ri, _r, _RS, R, _RR |
| 13531 | M | 0.15 | 0.97 | _MM, _m, MM, _mM, _mm, _mf, _ms, mM |
| 30 | T | 0.16 | 0.99 | _TT, _TC, TT, TC, _tc, _TG, _TS, _TD |
| 1761 | D | 0.14 | 0.98 | _DD, _D, _DS, _DP, _DT, DD, DP, DS, _Ds |
| 7342 | I | 0.13 | 0.91 | IB, i, IC, İ, IE, IH, IP, _IW, IR, IW |
| 2651 | U | 0.11 | 0.93 | _UA, U, _UT, UU, _U, _UF, _UD, UE, UA |
| 4664 | C | 0.14 | 0.93 | _getC, _CC, getC, _c, setC, CC, Cs, _Cs |
| 357 | B(+R) | 0.006 | 0.91 | _BR, _Br, Br, BR, _Bra, _br, Bra, br |
| 12114 | S(+L) | 0.006 | 0.95 | _SL, SL, _sl, _Sl, sl, Sl, _Slide |
| 14857 | C(+U) | 0.006 | 0.91 | _Cur, _cur, Cur, _CUR, cur, CUR, _Kur |

Table 2: **Examples of Alphabet Latents.** Latents from layer 25 of Gemma 2 2B that promote or suppress tokens sharing an initial letter. "Metric" is the fraction of top 100 affected tokens starting with that letter.

tokens [Land and Bartolo, 2024]. Motivated by prior work linking the $\mathbf{W}_U$ nullspace to an RMSNorm-based [Zhang and Sennrich, 2019] entropy regulation mechanism [Stolfo et al., 2024], we investigate whether these latents encode this internal computation.

To test whether these latents causally influence output entropy, we ablate the residual stream along each latent's decoder direction by setting its value to the corresponding decoder bias, thereby removing information in that direction. We then measure the change in per-token entropy of the model's output distribution. Figure 6b reports the entropy change for all latents with $\alpha_{10} > 0.3$ (one per antipodal pair to avoid redundancy), compared to a control group of 50 randomly selected latents.[7]

We find that some nullspace latents produce much larger entropy shifts than the random baseline, indicating that they encode signals relevant to entropy modulation. In particular, latent 14325 has a disproportionate impact on output entropy. To test whether this signal is used by the model in conjunction with RMSNorm scaling (as in Stolfo et al. [2024]), we repeat the ablation while freezing the RMSNorm scaling coefficient. Figure 6c shows that the entropy change diminishes under this intervention, suggesting that the model uses this direction to modulate entropy via RMSNorm. Furthermore, Figure 7 shows that the combined activation of the antipodal pair formed by latents 13748 and 14325 is strongly correlated with output entropy, further supporting this interpretation.

While these results highlight the functional role of specific nullspace latents in entropy regulation, not all latents in this subspace behave similarly. Some exhibit negligible impact on entropy when ablated. We speculate that these may track different internal signals–one such candidate is the attention sink signal, which has also been associated with the $\mathbf{W}_U$ nullspace [Cancedda, 2024]. Overall, these experiments provide mechanistic evidence that nullspace latents correspond to internal model computations.

---

[7]The entropy changes for the random latents are aggregated into a single boxplot.

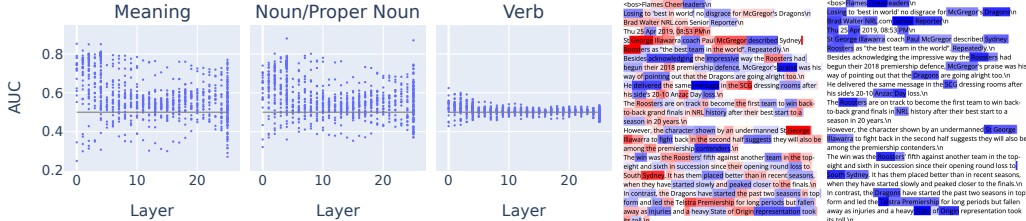

Figure 8: **Meaningful-Word Latents.** (Left) AUCs of predicting feature firing, from whether the POS tag is within the specific category. "Meaning word" and "noun/propernoun" are good predictors, while other categories like "verb" are less predictive. (Middle) Example of L2: pair 15089 (blue), 13092 (red) firing patterns on a document, where 15089 fires on "meaning-heavy" words while 13092 fires on proper nouns and functional words (the, in, a). (Right) Example of L3: 7507 firing patterns, where it fires selectively on proper nouns.

## 4.4 Alphabet Latents

We identify a class of dense latents that selectively boost or suppress large sets of tokens sharing the same initial letter. Unlike prior work that linked latents to the *current* token's first letter [Chanin et al., 2024], these instead relate to the *next* token's initial character.

To discover these latents systematically, we examine each latent's top 100 positive and negative logit contributions by projecting its decoder weights onto the vocabulary space. Then, we collect the corresponding tokens, and select latents where either set contains at least 90% of tokens starting with the same character (excluding the space character "_"). At layer 25, this procedure yields 114 such latents, of which 21 have activation density >0.1, accounting for 20% of all dense latents. These latents span a range of antipodality scores and activation densities, but notably appear as high-density features only at the model's final layer. We provide some examples from this layer in Table 2.

Interestingly, we observe multiple latents for each letter, varying in specificity: some target a broad set of short tokens sharing only the first letter (e.g., "b" or "c"), while others focus on longer tokens sharing a multi-letter prefix (e.g., "br" or "cu"). We attribute this granularity to feature splitting [Bricken et al., 2023] possibly driven by n-gram frequency, which yields latents with differing activation densities. These latents illustrate how SAEs dedicate dense units to encode output-specific signals related to next-token lexical structure.

## 4.5 Meaningful-Word Latents

The next class of latents that we investigate are those whose firing can be well predicted by the part-of-speech (POS) tag of the token. We create a reduced set of high-level tags from the Brown Corpus [Francis and Kučera, 1979] by combining similar tags (e.g., combining plural and singular forms of nouns),[8] and capture dense latent activations on 10k sentences ($\approx$ 200k tokens) from the corpus. Then, for each latent, we calculate the AUC-ROC of predicting the binary latent activations given the binary vector of whether a token is within the high-level POS category. Intuitively, this AUC reflects how well the interpretable linguistic category *predicts* the latent.

We find that even these high-level groupings are not enough to achieve a high AUC (Figures 8 and 20), and propose a further grouping of these tags into "meaningful words", where a token is considered a "meaningful word" if it is one of {nouns, proper nouns, verbs, adjectives, adverbs}. The resulting binary-binary predictor has a decent AUC (Figure 8) of $\approx$ 0.8 for many dense latents in early layers, suggesting that the model contains a dense subspace tracking the presence of these meaningful words.

## 4.6 PCA Latents

Since the top principal components (PCs) are a large fraction of the variance of the activations, one might expect an SAE to learn dense latents that simply reconstruct this subspace. However, we

---

[8]See Table 3 in Appendix E.6 for our full mapping.

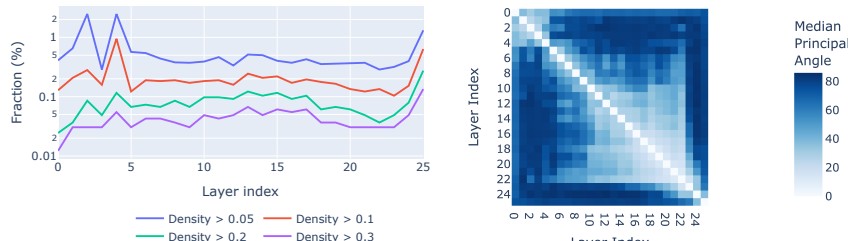

Figure 9: **Layer-wise Dynamics of Dense Latents.** (a) Fraction of dense latents (at various density thresholds) across residual stream SAEs at different layers of Gemma 2 2B. (b) Median principal angles between dense-latent subspaces, showing a shift in subspace structure from early to late layers.

find that this hypothesis is only partly the case: as shown in Figure 21, an antipodal pair of latents consistently reconstruct most of the first PC (cosine similarity $> 0.75$), but other latents do *not* have a large norm percentage in the top PC, even up to the top 5 PC components. The top PC-aligned latents are generally not immediately interpretable and do not fall into any of our classes above. Interestingly, decreasing or increasing the SAE $L_0$ and dictionary size does not eliminate PC-aligned latents nor result in significantly more of them (Figure 22).

### 4.7   Layer-wise Dynamics

As noted in the taxonomy of dense latents above, and visualized in Figure 3, each class of dense latents is found in specific layer ranges. Dense latents in early layers have more token-dependent activations and track positional information, those in middle layers represent more conceptual directions, and those in the final layers are mostly mechanisms that the model uses to control its output. Inspired by these observations, in this section we further examine layer-wise characteristics of dense latents.

**Number of Dense Latents.**   First, we study how the number of dense latents changes across different layers of the model. Figure 9a illustrates the fraction of latents exceeding density thresholds of 0.05, 0.1, 0.2, and 0.3 at each layer. In the early layers (0-4), we observe transient spikes in latents just above the 0.05 and 0.1 thresholds. These latents are largely the part-of-speech related latents in §4.5. The absence of similar spikes at the 0.2 and 0.3 thresholds suggest that these early fluctuations arise from SAE training variability rather than fundamental differences in the information encoded at different points of the model's residual stream. Across the middle layers (5–23), the fraction of dense latents is remarkably stable for all thresholds. Finally, the model's last two layers exhibit an increase in the number of dense latents, indicating a final emergence of dense features prior to unembedding.

**Consistency of the Dense Subspace.**   We next ask whether the subspace spanned by dense latents remains stable across layers or varies over the model. For each pair of layers, we compute the principal angles between the subspaces defined by latents with density $> 0.2$, then take the median angle as a summary statistic: values near $0°$ indicate largely overlapping subspaces, while values near $90°$ indicate dissimilarity. Figure 9c visualizes these median angles for every layer pair of Gemma 2 2B.[9] Three clusters emerge. Layers 0-4 share a common dense subspace (low angles). This shifts in the middle of the model (layers 10–22), where a new stable subspace persists (mutually low angles). Finally, the last few layers exhibit a pronounced change (large angles relative to earlier layers), consistent with the rise of alphabet and nullspace latents before the unembedding.

## 5   Related Work

**Sparse Autoencoders.**   Transformer models are thought to represent features as linear directions in activation space [Mikolov et al., 2013, Bolukbasi et al., 2016, Elhage et al., 2021, Nanda et al., 2023, Park et al., 2024, Olah, 2024], with many more features than neurons, leading to *superposition* [Olah et al., 2020, Elhage et al., 2022]. Early work explored sparse dictionary learning to interpret

---

[9]We find that using a slightly higher density threshold (0.2) makes the subspace similarity pattern more pronounced. The same plot with a lower threshold (0.1) is shown in Appendix A.3, showing the same clustering trend but with reduced overall similarity.

these representations [Olshausen and Field, 1997, Faruqui et al., 2015, Arora et al., 2018, Zhang et al., 2021]. More recently, sparse autoencoders (SAEs; Ng et al., 2011) have emerged as a scalable and effective implementation of sparse dictionary learning for transformer-based models [Yun et al., 2021, Bricken et al., 2023, Huben et al., 2024, Rajamanoharan et al., 2024a,b, Kissane et al., 2024, Bussmann et al., 2025] that can recover meaningful and causally important features [Templeton et al., 2024, Gao et al., 2025, Marks et al., 2025].

**Interpreting SAE Latents.** As SAEs have gained traction, recent work has focused on interpreting the meaning of their latent features [Chanin et al., 2024, Leask et al., 2025]. Building on the neuron interpretation methodology in [Bills et al., 2023], several recent works interpret SAE latents systematically. Templeton et al. [2024] propose a rubric-based evaluation method in which a language model (Claude 3 Opus) scores how well a proposed feature description aligns with the contexts on which the latent activates. Similarly, Paulo et al. [2024] propose a pipeline in which natural language interpretations for SAE latents are matched with different contexts and used by an LLM in different tasks that evaluate how good the interpretations are in predicting activating and non-activating contexts. Other recent efforts explore automated interpretation approaches based on self-interpretation strategies [Kharlapenko et al., 2024]. A recurring observation across multiple studies is *dense* latents, which activate on more than 10% or even 50% of tokens [Cunningham and Conerly, 2024, Rajamanoharan et al., 2024b]. Chen and Batson [2025] take the 10 most densely activating latents in a cross-layer Transcoder trained on Claude and attempt to manually interpret them, finding plausible interpretations (e.g., "activates on commas," "activates on non-terminal tokens in multi-token words") for 6 of the 10 features. In contrast, Rajamanoharan et al. [2025] view dense latents as an undesired phenomenon and propose a frequency-based regularizer to discourage their emergence during training. Whether these latents reflect meaningful internal computations or arise as undesirable artifacts was up until our work an open question.

**Dense Language Model Representations.** Prior work has also identified dense signals in language model representations more broadly (i.e., components that encode information consistently across many tokens). Gurnee et al. [2024] present a taxonomy of universal neurons that appear across GPT-2 models trained with different seeds. Among these, they identify neurons that encode positional information. Chughtai and Lau [2024] similarly identify dense positional features in an SAE trained on GPT-2's layer 0, though they do not explicitly analyze their activation density. Finally, Stolfo et al. [2024] describe neurons that regulate model confidence by tracking entropy and connect them to a component of the residual stream aligned with the quasi-nullspace of the unembedding matrix.

## 6 Discussion, Limitations & Conclusion

Our work shows that dense SAE latents discover intrinsically dense features in the underlying language model representations. This challenges recent efforts that aim to remove dense latents with ad-hoc penalties in the SAE loss function [Rajamanoharan et al., 2025]. Our results motivate future feature-extraction mechanisms that are able to find features that are not necessarily sparse. For example, such techniques might include SAE designs that allocate autoencoder capacity for representing dense subspaces, approaches that optimize circuit sparsity, or techniques like APD [Braun et al., 2025] that focus on parameter sparsity.

**Limitations.** Although our work identifies some classes of dense latents, we do not claim that all dense latents encode interpretable or meaningful signals. We hypothesize that some dense latents are a noisy aggregation of sparse features rather than a "true" dense feature, and distinguishing between these remains an open challenge. Moreover, dense latents may learn a basis that spans *but does not align with* the set of true dense model representations, since dense latents co-occur extremely frequently, and a linear combination of the "true" basis works for reconstruction too.

Despite consistently observing the antipodality trend across both TopK and JumpReLU SAEs and across models (Gemma 2 2B and GPT-2 Small), our interpretability analysis primarily focuses on JumpReLU SAEs trained on Gemma 2 2B, using a single dictionary size and sparsity constraint per layer. Future work could broaden analysis to more models, SAE architectures, and SAE sparsities.

Most notably, we have explained less than half of dense SAE features. We view understanding the rest of these latents as exciting future work that could provide insight into frequently-active, fundamental mechanisms and representations in language models.

## Acknowledgments

We would like to express our gratitude to Arthur Conmy, Neel Nanda, and Vilém Zouhar for their valuable feedback and insightful discussions at different points during the development of this project. AS acknowledges the support of armasuisse Science and Technology through a CYD Doctoral Fellowship.

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

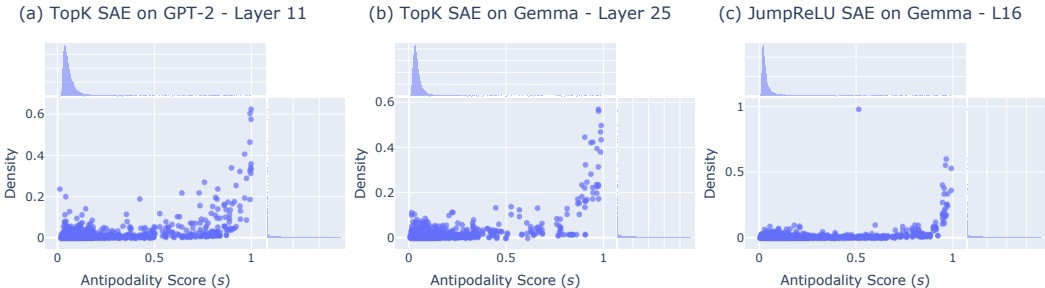

Figure 10: **Additional Antipodality Plots.** Antipodality scores vs. activation density for (a) TopK SAE on GPT-2 (Layer 11), (b) TopK SAE on Gemma 2 2B (Layer 25), and (c) JumpReLU SAE on Gemma 2 2B (Layer 16). Across all configurations, dense latents tend to have high antipodality scores.

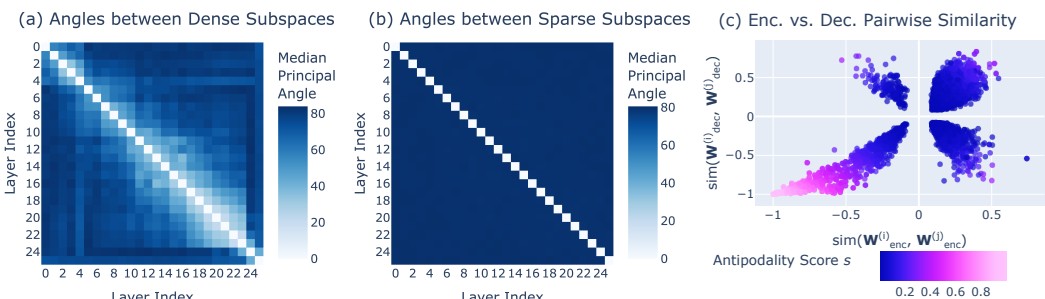

Figure 11: **Additional Analyses.** (a) Median principal angles between dense-latent subspaces (density>0.1) across layers. (b) Principal angles between randomly selected non-dense latent subspaces. (c) High antipodality score occurs when encoder and decoder weights are nearly opposite.

## A  Additional Results

### A.1  Antipodal Pairing in Different SAEs

Figure 10, we report antipodality scores (computed as in Eq. (1)) for dense latents in three additional SAEs: two TopK SAEs that we trained on the residual streams of GPT-2 (layer 11) and Gemma 2 2B (layer 25), and a JumpReLU SAE from the Gemma Scope suite trained on an earlier layer (16). In all cases, we observe the same trend highlighted in Figure 1c: high-density latents cluster at high antipodality scores, forming near-antipodal pairs that reconstruct specific directions in residual space.

### A.2  Dense Latents During Training

In Figure 12, we visualize the number of dense latents (activation frequency $> 0.1$) over training steps for each SAE configuration in our ablation experiment described in §3.1. All curves converge within the first ~100k steps and remain stable throughout training. This early plateau suggests that dense latents are not a product of late-stage optimization noise, but rather emerge early and persist, indicating that they reflect consistent structure in the residual stream rather than transient artifacts.

### A.3  Angles Between Residual Stream Subspaces

In Figure 11, we provide further analysis of the evolution of dense latent subspaces across layers. Panel (a) shows the median principal angle between the subspaces spanned by latents with density $> 0.1$ at each pair of layers in Gemma 2 2B. These results follow the trend observed in Figure 9c (based on a $> 0.2$ cutoff),

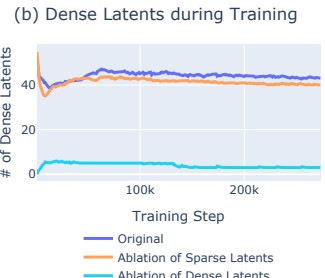

Figure 12: **Dense Latents During Training.** Dense latent counts stabilize early in training.

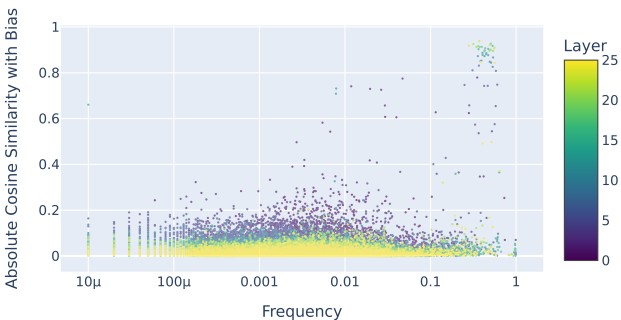

Figure 13: Plot of absolute cosine similarity of all SAE decoder vectors at all layers with that layer's decoder bias. We observe a group of dense latents in the upper right corner that have high frequency and align with the bias.

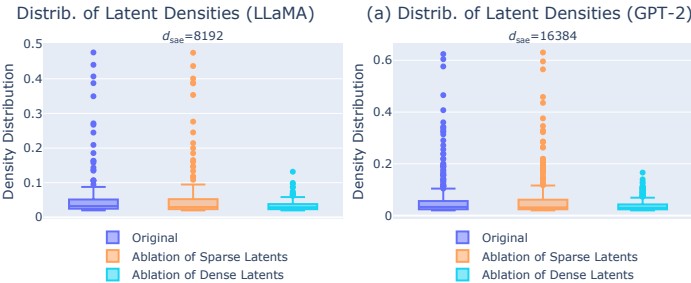

Figure 14: **Dense-subspace ablations on LLaMA-3.2-1B and GPT-2 Small.** For each model's final layer we train a baseline SAE (blue), retrain after ablating the subspace spanned by dense latents (teal), and retrain after ablating an equally sized subspace of non-dense latents (orange). Only removing the dense-latent subspace collapses the high-density tail.

revealing distinct subspace clusters in the early, middle, and late layers. However, the overall similarity between subspaces is lower here, reflecting the greater variability introduced by including moderately dense latents (density 0.1-0.2).

For comparison, panel (b) reports the same metric computed on subspaces spanned by 100 randomly selected non-dense latents per layer. As expected, these subspaces exhibit minimal overlap, with median principal angles near 90° across all layer pairs, confirming that the structure observed in the dense-latent subspaces is nontrivial.

### A.4 Pairwise Similarity Between Latents' Weights

In Figure 11c, we report for each latent $i$, the maximum-magnitude cosine similarity of its encoder and decoder weights with any other latent $j$. In particular, we show $\text{sim}(\mathbf{W}_{\text{enc}}^{(i)}, \mathbf{W}_{\text{enc}}^{(j)})$ and $\text{sim}(\mathbf{W}_{\text{dec}}^{(i)}, \mathbf{W}_{\text{dec}}^{(k)})$, where $j = \arg\max_{l \neq i}(|\text{sim}(\mathbf{W}_{\text{enc}}^{(i)}, \mathbf{W}_{\text{enc}}^{(l)})|)$ and $k = \arg\max_{l \neq i}(|\text{sim}(\mathbf{W}_{\text{dec}}^{(i)}, \mathbf{W}_{\text{dec}}^{(l)})|)$. We find that the antipodality score $s$ approaches 1 only when both encoder and decoder similarities are close to $-1$.

### A.5 Similarity with SAE Bias

To investigate the relationship between dense latents and the SAEs' bias terms, we compute the cosine similarity between each SAE decoder vector and the corresponding layer's decoder bias. Figure 13 shows the absolute cosine similarity for all latents across layers as a function of activation frequency. We observe a small but distinct group of dense latents (upper-right region) that strongly align with the bias.

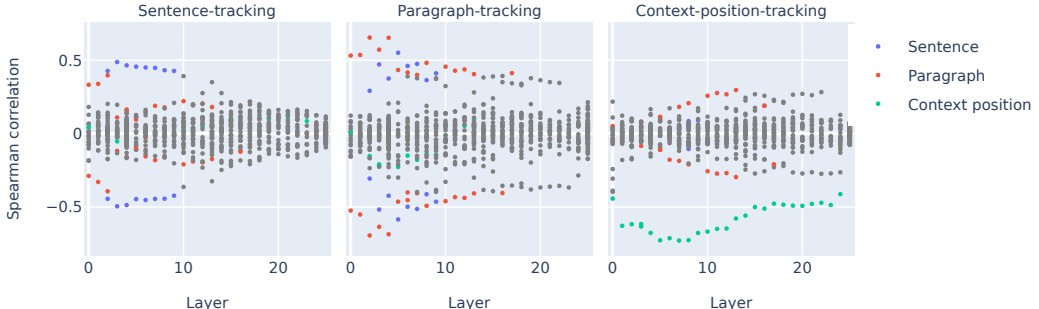

Figure 15: **Position Latents**. We identify position latents by computing their Spearman correlation $\rho$ with relevant text boundaries. We classify a latent as belonging to a certain category when $|\rho| > 0.4$.

## A.6 Additional Dense-latent Subspace Ablations

Figure 14 replicates the dense-subspace ablation from §3.1 on two additional models: LLaMA-3.2-1B [AI @ Meta, 2024] and GPT-2 Small [Radford et al., 2019]. For each model we (i) train a baseline SAE, (ii) retrain after zero-ablating the subspace spanned by dense latents, and (iii) retrain after ablating an equally sized subspace of randomly chosen non-dense latents. All experiments are run at the final residual-stream layer. In both cases, removing the dense-latent subspace collapses the high-density tail—yielding almost no dense latents—whereas ablating a sparse subspace leaves the distribution essentially unchanged. These replications mirror the Gemma 2 2B result and further support that dense latents reflect an intrinsic residual-stream subspace rather than a training artifact.

## B Experimental Details

For the experiment in §3.1, we trained TopK SAEs [Gao et al., 2025] on the residual stream activations at layer 25 of Gemma 2 2B using 1 billion tokens from the OpenWebText corpus [Gokaslan and Cohen, 2019]. Training followed the default configuration of the `Sparsify` library,[10] and experiment tracking was conducted using Weights & Biases.[11] The ablation experiment on nullspace latents described in §4.3 was performed on a 10k-token subset of the C4 corpus [Raffel et al., 2020]. Analyses throughout the paper were conducted using the Gemma Scope SAEs [Lieberum et al., 2024] with 16k latents trained on the residual stream of Gemma 2 2B. All experiments were implemented in `PyTorch` [Paszke et al., 2019], with model inspection tools from the `TransformerLens` library [Nanda and Bloom, 2022]. Data processing used `NumPy` [Harris et al., 2020] and `Pandas` [Wes McKinney, 2010], and figures were generated with `Plotly` [Plotly Technologies Inc., 2015].

## C Compute Resources Used

We expect the experiments for training SAEs, capturing SAE activations and generating completions with Gemma 2 2B to be able to be run in about 30 A6000 hours. The LLM judging experiments take less than USD $20 through OpenRouter with Gemini 2.5 Flash Preview [Gemini Team, 2025].

## D Broader Impact

Our work focuses on interpreting language models, an important component of building safer and more reliable systems. SAEs in particular are a popular technique for understanding language models, and through investigating dense latents, we can both better inform SAE design, and better understand language model internals.

We do not foresee any negative impacts of our work.

---

[10]`https://github.com/EleutherAI/sparsify`
[11]`https://wandb.ai`

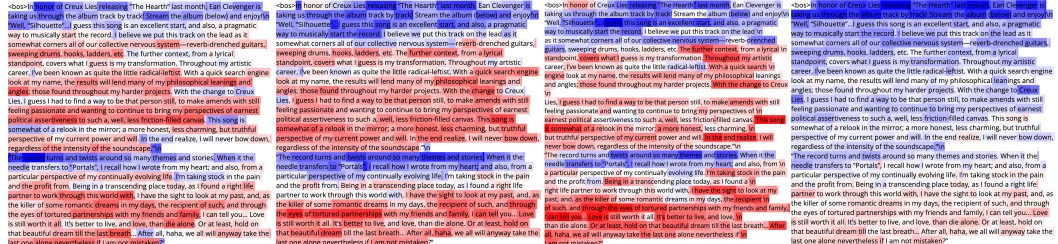

Figure 16: Examples of position latents in layer 5. Deep blue represents positive projection along decoder vector, and deep red represents negative. (1) L5:4341 is a sentence-tracking latent, that lights up consistently on beginnings of sentences. It has strong activations for topic sentences too. (2) L5:8680 is a paragraph-tracking latent, that lights up on beginnings of paragraphs. (3) L5:8680 is agnostic to artificially adding formatting newlines, showing it is encoding true paragraph position. (4) L5:697 is a context-position-tracking latent.

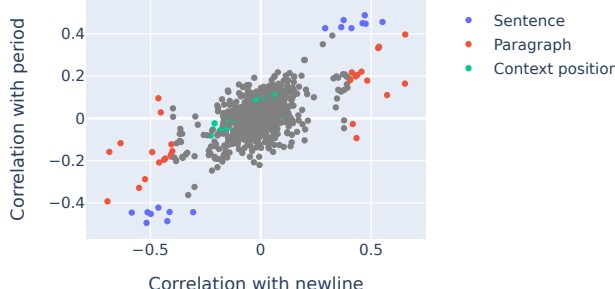

Figure 17: Spearman correlation for period against Spearman correlation for newline.

# E  Additional Taxonomy Results

## E.1  Classification of dense latents

In our taxonomy, we identify dense latents using automated tests. We do not expect these tests to be perfect for a variety of reasons—for instance, dense latents not lining up perfectly with the "true" feature basis due to learning a linear combination basis, and the fundamental difficulty of designing true, causal tests. However, for the purposes of illustration, we choose reasonable cutoffs for each test to create Figure 3, listed below.

- Position latents: Spearman correlation of $|\rho| > 0.4$ for the relevant text boundary.
- Context-binding latents: Fraction of successful flips $> 0.75$.
- Nullspace latents: $> 0.2$ of encoder weight in bottom 10 $\mathbf{W_U}$ singular vector subspace.
- Alphabet latents: Top 100 or bottom 100 logit contributions contain at least 90% of tokens starting with same character.
- Meaningful-word latents: AUC of using "is meaningful word" to predict "feature fires" $> 0.75$.
- PC-aligned latents: cosine similarity with top PC $> 0.75$.

Very few dense latents (3.6% across layers) fall in >1 category based on our automated tests to find them, with the most common clashes being between sentence- and paragraph- tracking (see Appendix E.2), and between several categories and meaningful-word latent. For the purposes of illustration, we break ties according to the priority (from highest to lowest): {context-tracking, sentence-tracking, alphabet, nullspace, context-binding, paragraph-tracking, meaning, PCA} based on our confidence in our automated tests.

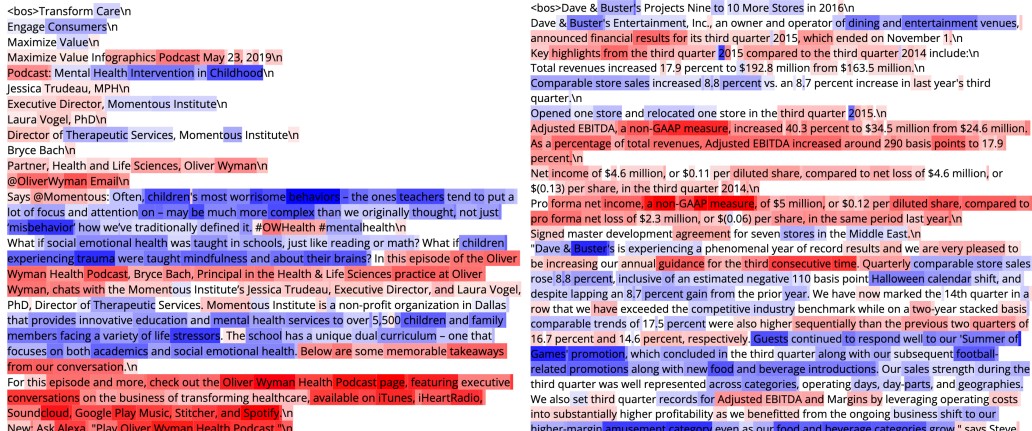

Figure 18: L13: 15275 (blue) and 11449 (red), which has 81.5% correct flips. In these two examples, 15275 fires on children's mental health (left) and Dave & Buster's promotions (right), while 11449 fires on mentions of the podcast (left) and financial measures (right).

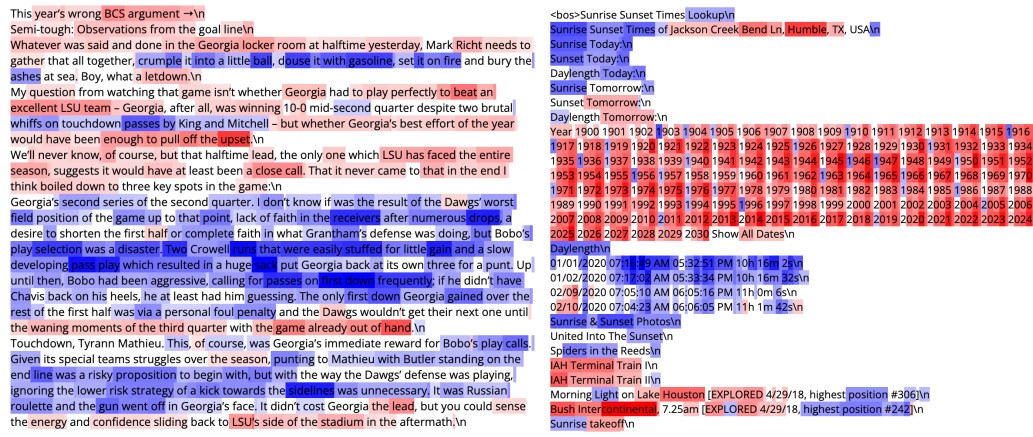

Figure 19: L12: 14906 (blue) and 14599 (red), which has 76.5% correct flips. In these two examples, 14906 fires on descriptions of the game (left), and text or numbers related to sunrise (right), while 14599 fires on the teams and winning/losing (left), and years or locations (right).

## E.2 Position latents

The observation in Figure 17 that period-tracking and newline-tracking latents are hard to distinguish also relates to our discussion in §6 that because the sparsity incentive is low for these dense latents, they may not be perfectly aligned to "true" model dense features, and may instead be a linear combination of two related features.

## E.3 Interpreting context-binding latents

We attempt to interpret mid-layer latents that exhibit coherent chunk-level activations in two ways:

1. **All-context**: Following existing autointerp methods [Paulo et al., 2024], we sample 10 activating and 10 non-activating phrases from an entire corpus and ask an LLM (Gemini 2.5 Flash) to generate an explanation. We repeat this 100 times to generate 100 explanations.

2. **In-context**: We instead sample 10 activating and 10 non-activating phrases from the *same context*. We repeat this 100 times (using 100 different contexts) to generate 100 explanations.

When examples are drawn from the same context, the explanations are specific but highly diverse across contexts. When examples are drawn from different contexts, the explanations become vague or

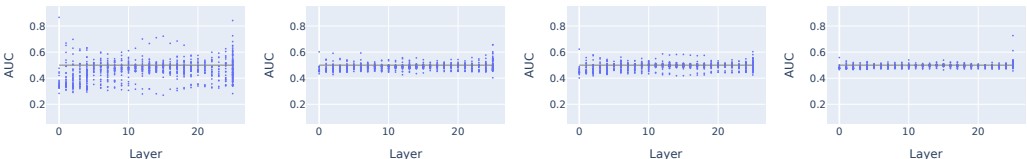

Figure 20: From left to right, we show the AUCs of predicting latent firing using function words (any of {'article', 'prepos', 'conjunction', 'det', 'modal', 'be', 'do', 'have', 'what'}), articles, prepositions and conjunctions. These do not do as well as the "meaningful-word" or "noun/propernoun" groupings.

| Category | Tags |
|---|---|
| punc | .  (  )  *  -  ,  :  " "  ' |
| quantifier | ABL, ABN, ABX, AP, AP$ |
| article | AT |
| be | BE, BED, BEDZ, BEG, BEM, BEN, BER, BEZ |
| conjunction | CC, CS |
| num | CD, OD |
| do | DO, DOD, DOZ |
| det | DT, DTI, DTS, DTX, DT$ |
| have | HV, HVD, HVG, HVN, HVZ |
| prepos | IN, TO |
| adj | JJ, JJR, JJS, JJT |
| modal | MD |
| noun | NN, NN$, NNS, NNS$, NR, NRS, NR$, UH |
| propernoun | NP, NP$, NPS, NPS$ |
| pronoun | PN, PN$, PP$, PP$$, PPL, PPLS, PPO, PPS, PPSS |
| qual | QL, QLP |
| adv | RB, RB$, RBR, RBT, RN, RP |
| verb | VB, VBD, VBG, VBN, VBZ |
| what | WDT, WP$, WPO, WPS, WQL, WRB, EX |
| unknown | NIL |

Table 3: Mapping from high-level category to Penn Treebank tags. A trailing $ marks possessive forms.

generic (Table 4). This drop in specificity across contexts is somewhat expected, since explanations for any latent may overfit the context. However, doing the same for sparse latents (Table 5), we see that a "good" sparse latent would have similar explanations with both in-context and all-context examples, aligning with the usual assumption that SAEs learn directions that represent a concept in the model.

It is difficult to rule out the possibility that these dense latents represent an uninterpretable abstract feature the model learns. However, the steering experiment seems to cause the relevant specific concepts to be brought up during generation, supporting the "binding" hypothesis that there are dense directions that do not represent a fixed concept but rather are used in the model's computation.

### E.4 Additional examples of context-binding latents

We include two additional examples of context-binding latent pairs with high flip score: layer 13 pair (15275, 11449) (Figure 18) and layer 12 pair (14906, 14599) (Figure 19). For each pair, we show two example contexts where they are active, illustrating how each latent activates on specific but context-dependent concepts, and that latents in a pair do not co-activate.

### E.5 Steering context-binding latents

Our methodology for steering is as follows:

1. Prompt Gemma 2 2B with input text from the RedPajama dataset, ending at a natural point (after a newline token), with at least 400 tokens.

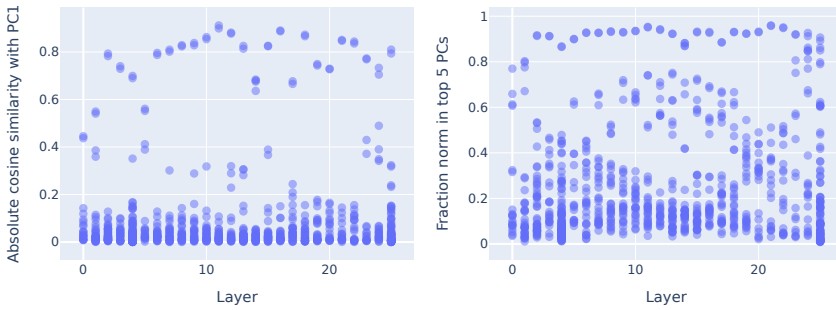

Figure 21: (Left) Cosine similarity of dense latents with top principal component. (Right) Fraction norm of dense latents in top 5 principal components.

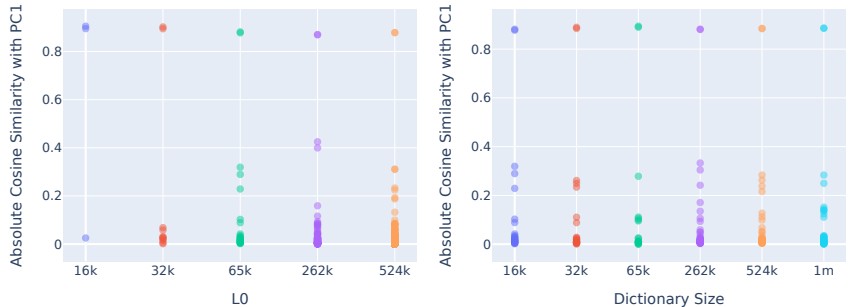

Figure 22: Cosine similarity of dense latents in layer 12 with the top principal component, across different L0s and SAE dictionary sizes.

2. Capture the activating phrases of F1 and F2 that are at least 5 consecutive tokens long.

3. Allow Gemma 2 2B to generate a completion without steering, and prompt an LLM (Gemini 2.5 Flash Preview) to judge whether the completion is more like F1 activating examples, F2 activating examples, or unclear.

4. Repeat the above, but steering on the last token during generation, in the direction of F1 and F2. Since F1 and F2 are antipodal pairs, we first ablate the subspace spanned by F1 and F2, before adding the steering vector, that is fixed at 2x the historical activation of that feature in that context.

### E.6 Meaningful-Word Latents

We provide the mapping from higher-level categories to Penn Treebank tags in Table 3. In addition to the AUC for the categories shown in the text, we show the AUC for a few other categories in Figure 20, which seem mostly unrelated to predicting the activations of dense latents.

### E.7 PCA latents

Figure 21 reports, layer-by-layer, each dense latent's cosine similarity to PC1 (left) and the share of its norm contained in the top-5 PCs (right). As noted, only one antipodal pair per layer strongly aligns with PC1; most dense latents place little mass in the top PCs. Figure 22 shows that this pattern is stable under SAE hyperparameters: varying the sparsity target ($L_0$) or dictionary size yields similar PC1 alignments, neither removing nor proliferating PC-aligned latents.

## Dense latent L12:7541 ($f = 0.421$)

| In-context | All-context |
|---|---|
| Numerical values or categories related to real estate listings, such as prices, property types, or number of beds | A proper noun or a specific concept/entity within a sentence |
| Mentions of people, groups, or entities involved in a discussion or action | The beginning of a new clause or phrase, often following a comma, parenthesis, or other punctuation, or indicating a shift in topic or focus |
| Mentions of Donna Brazile or political news and commentary, often critical of establishment figures or media outlets | A sequence of words that are part of a proper noun, a title, or a specific phrase, often capitalized, or a short phrase that acts as a label or identifier within a larger text. |
| References to the Niagara Wire publication or its staff and content | The beginning of a new sentence or clause, often following punctuation or a line break |
| Terms and phrases related to the Cybercrime Prevention Act of 2012 in the Philippines | The detection of numbers, dates, or specific numerical references within text |
| References to Barack Obama or his administration | The beginning of a new clause or phrase, often following a conjunction, preposition, or punctuation mark, that introduces additional information or a new element into the sentence structure |
| Mentions of specific people, places, or things, or references to a particular time or event | A proper noun or a common noun that is part of a larger phrase, often appearing after a preposition or as an object of a verb |
| Mentions of CSIRO's internal operations, expertise, and collaborative efforts | Mentions of specific entities, objects, or concepts within a broader context, often highlighting a particular detail or aspect of the surrounding text |
| Nautical vessel chartering and related services | The letter 'I' or 'A' or 'O' when it is the first letter of a word or a standalone word |
| Mentions of healthcare organizations, roles, or initiatives related to health and human services | A token or sequence of tokens that is part of a larger, multi-word proper noun, compound noun, or specific phrase, where the preceding context helps to complete the meaning of the highlighted part. |

## Dense latent L12:2009 ($f = 0.319$)

| In-context | All-context |
|---|---|
| Punctuation marks, numbers, or single letters that are not part of a larger word | Code, symbols, or foreign language phrases |
| Mentions of events, dates, or outcomes related to the life of Stéphanie of Monaco | The continuation of a word or phrase across a line break or other formatting boundary |
| Mentions of the St. John's Red Storm basketball team, their coach Mike Jarvis, or their player Hatten | Short, common words or symbols that are often part of a larger phrase or structure, but do not carry significant meaning on their own |
| Reporting on COVID-19 cases and related news | The beginning of a new sentence or clause, often following a period, comma, or other punctuation, or a line break |
| Scientific citation formatting and punctuation | Punctuation marks, prepositions, and conjunctions that connect different parts of a sentence or list |
| The beginning of a new clause or sentence | The beginning of a new word or token that is not preceded by a space |
| References to students or pupils in an educational context | The detection of a word or phrase that is part of a larger, well-known entity or common expression, where the detected part is not the beginning of the entity or expression |
| Biographical details and life events of a character, including family, career, and personal status, often presented in a chronological or list-like format | Mentions of specific people, places, or entities, or phrases that introduce or refer to them |
| Specific references to the current or a past UN General Assembly session, or to the UN Secretary General and his staff | The beginning of a new sentence or phrase, often following punctuation or a line break, or the start of a new section within a document |
| Mentions of specific dates, years, or numbers in a historical or official context | Mentions of specific words or phrases that are part of a larger list or enumeration, often found in titles, bullet points, or structured content |

Table 4: Sampled explanations of dense latents L12:7541 and L12:2009, using in-context examples versus all-context examples. The in-context explanations are highly specific and diverse, while the all-context explanations are vague.

## Sparse latent L12:10356 ($f = 8.90 \times 10^{-4}$)

| In-context | All-context |
|---|---|
| Years in the 2010s or 2020s, often following a movie title and sometimes preceded by "HBO" or "HBO Max" | A two-digit year following a month or day, or as part of a date range |
| The number "1" in a four-digit year, specifically in the 2010s decade | The last two digits of a four-digit year |
| The year 2019 in dates or as a standalone year | The last two digits of a four-digit year |
| The digit '1' when it is part of a four-digit year that starts with '20' and is followed by a two-digit number, typically representing a day or a time, or a forward slash. | The second digit of a year in the 21st century |
| The first digit of a two-digit year in a citation | The last two digits of a four-digit year |
| The first two digits of a four-digit year | The last two digits of a year in the 21st century |
| The release year of a movie title | The last two digits of a four-digit year |
| Mentions of the "Product of the Year" award followed by a specific year | The last two digits of a year in a date |
| The year 2022 in date formats | A four-digit year in the 21st century, specifically between 2010 and 2024 |
| The third digit of a four-digit year, specifically when the year is 2013, often found in movie titles or release dates | The last two digits of a four-digit year, typically in the 2000s |

## Sparse latent L12:800 ($f = 7.30 \times 10^{-4}$)

| In-context | All-context |
|---|---|
| Mentions of the South Ossetian conflict, including locations, people, and related events | Mentions of people's names or titles, often followed by their statements or actions |
| References to a specific person named Sarah, including possessive forms and direct address | Mentions of a person or entity speaking or being referenced |
| Mentions of reports, analyses, or statements made by individuals or groups | A proper noun or pronoun that is the subject of a sentence or clause, or a proper noun that is the object of a verb or preposition |
| Mentions of individuals or organizations involved in mine clearance or humanitarian aid | Mentions of people or organizations, often in attribution or as subjects of actions |
| Mentions of the author Tom Bissell, often in relation to his work or statements | Mentions of proper nouns, often names of people or organizations, that are split across a line break |
| Mentions of "Dicko" as a proper noun, often followed by a verb or punctuation, indicating a new clause or action related to the person. | Mentions of people speaking or being quoted |
| Mentions of Alan Waller, Earl Spencer's former head of security | Proper nouns or pronouns referring to people or organizations, often followed by a verb |
| Mentions of people or organizations as subjects or possessive entities | Mentions of a person's name followed by a verb of speaking or a reference to that person |
| Attributions of statements or opinions to individuals or groups, often experts, in news or analytical contexts | A proper noun or pronoun that is the subject of a sentence or clause |
| Mentions of Sonny Dykes, a football coach, or his last name, often in the context of his statements or actions | Mentions of people or organizations as subjects or agents of actions |

Table 5: Sampled explanations of sparse latents L12:10356 and L2:800, using in-context and all-context examples. While the in-context explanations still tend to be more specific, they still center around a similar theme as the all-context explanations, and it is plausible that L12:10356 is a "date" feature and L12:800 is a "proper noun" feature.

