# OpenReview forum: "Dense SAE Latents Are Features, Not Bugs"
_NeurIPS.cc/2025/Conference — NeurIPS 2025 poster_

### Official Review · Reviewer_59xx · 2025-06-25

**Clarity:** 2
**Significance:** 3
**Originality:** 4
**Rating:** 4
**Confidence:** 2

**Summary:**

This paper demonstrates that densely activating SAE latents are not mere training artifacts but correspond to principled directions in a model's residual-stream subspace. It shows that many of these latents come in antipodal pairs that encode a single one-dimensional feature, maps how dense latents bind to multiple contextual signals including context-binding units and introduce new taxonomy of the dense latents.

**Questions:**

- In Section 3.1, how exactly are the dense-latent components removed before retraining the SAE?
- Because the taxonomy relies on a properly trained SAE, could under- or over-fitting of the autoencoder bias the classification? Wouldn’t a direct statistical analysis of the raw LLM activations avoid this dependency?
- Are there latents that play more than one role, for example, units that behave both as position latents and context-binding latents?
- PCA latents are reported, but what interpretive value do they have—how should we understand their function?

**Ethical Concerns:**

["NO or VERY MINOR ethics concerns only"]

**Final Justification:**

Concerns remain regarding biases arising from the training process and the autoencoder, and the quantitative evidence is not fully sufficient. Nevertheless, I consider the study to be highly original and will therefore maintain the current score.

**Limitations:**

yes

**Paper Formatting Concerns:**

No major formatting issues.

**Quality:**

3

**Strengths And Weaknesses:**

## Strengths
- The authors provide *quantitative evidence* that dense latents reflect the geometry of the residual-stream subspace rather than incidental training artifacts.
- The paper carefully analyzes the antipodal-pair phenomenon in SAEs, showing how two latents encode positive and negative versions of the same direction.
- The study quantifies how strongly dense latents correlate with multiple contexts, revealing their broader representational role.
- The authors identify context-binding latents that preserve the current topic and statistically characterize their antipodal behavior.
- The work shows that null-space latents function like an internal temperature control, a novel and intriguing finding.

## Weakness
- Generality remains unclear: with experiments limited to Gemma 2, the findings might still be model-specific. Replicating the core result, especially the claim that dense latents are not SAE artifacts, on several other LLMs would strengthen the paper.
- The main text does not fully explain how each latent class is assigned. A unified, detailed description of the classification criteria (ideally consolidated in the appendix) is needed, including justification for heuristic thresholds such as the chosen Spearman $\rho$ cut-offs.

---

> ### Author Rebuttal · Authors · 2025-07-31
>
> Thank you for your valuable feedback and for acknowledging the “novel and intriguing finding” of our work.
>
>  > **W1:** Generality remains unclear: with experiments limited to Gemma 2, the findings might still be model-specific. Replicating the core result, especially the claim that dense latents are not SAE artifacts, on several other LLMs would strengthen the paper.
>
> To address this point, we replicated the dense subspace ablation experiment from Section 3.1 on a second model, GPT-2. As with Gemma 2, we find that training an SAE on activations with the dense subspace ablated results in almost no dense latents, while ablating an equally-sized sparse subspace has no comparable effect. These additional results strengthen the evidence that dense latents are not artifacts of SAE training and generalize beyond a single model. We also note that Appendix A.1 already includes a replication of the antipodality results on GPT-2. We will include these additions in the updated version of the paper.
>
> > **W2:** The main text does not fully explain how each latent class is assigned. A unified, detailed description of the classification criteria (ideally consolidated in the appendix) is needed, including justification for heuristic thresholds such as the chosen Spearman cut-offs.
>
> We detailed in Appendix E.1 the tests and cutoffs we used for each category of dense latents. We chose heuristic thresholds by identifying latents with outlier scores for the corresponding test (e.g. for Spearman cut-offs, see Figure 13 in Appendix E.2; for AUC cut-offs, see Figure 19 in Appendix E.6), and estimating a reasonable cut-off. We acknowledge that there is some noisiness to choosing fixed cut-offs in this way, and we do not claim that our classification is fully accurate but serves more as a guide. We will improve the clarity of the consolidated table in the final version, with references to the corresponding plots.
>
> > **Q1:** In Section 3.1, how exactly are the dense-latent components removed before retraining the SAE?
>
> To ablate this subspace from the language model’s residual stream, we remove the component of the residual that lies within it. Formally, let $\mathbf{W} \in \mathbb{R}^{k \times d}$ be the matrix of the latents’ decoder weights and $x \in \mathbb{R}^d$ the residual stream. We compute the projection $\mathbf{p} = \mathbf{x} \mathbf{W}^* \mathbf{W}$ (where  $ \mathbf{W}^*$ is the pseudoinverse of $\mathbf{W}$) and remove it: $\mathbf{x}’ = \mathbf{x} - \mathbf{p}$. This yields the ablated residual stream $\mathbf{x}’$. We will add an appendix section to clarify this procedure in the final version of the paper.
>
> > **Q2:** Because the taxonomy relies on a properly trained SAE, could under- or over-fitting of the autoencoder bias the classification? Wouldn’t a direct statistical analysis of the raw LLM activations avoid this dependency?
>
> This is a good question. We actually do study how the number of dense latents changes across training (Appendix A.2), and we observe that the number of dense latents stabilize early in training. Therefore, we do not expect that they are a result of under-/over- fitting.
>
> As for the direct statistical analysis of the raw LLM activations, dictionary learning is necessary to isolate and identify the directions that are frequently present in the activations, and it is unclear how we would otherwise find the “dense” subspaces from the LLM activations alone.
>
> > **Q3:** Are there latents that play more than one role, for example, units that behave both as position latents and context-binding latents?
>
> In our analysis, we did not find clear evidence of latents that play multiple roles. Due to the nature of our classification using the chosen heuristics and cutoffs, there are a few (3.6%) dense latents that fall into more than one category. Most overlaps are between sentence- and paragraph- tracking latents (as discussed in Appendix E.2, Figure 15), as well as between meaningful-word and nullspace/alphabet latents (likely due to the cutoffs chosen).
>
> > **Q4:** PCA latents are reported, but what interpretive value do they have—how should we understand their function?
>
> We were unable to form hypotheses on interpreting PC-aligned latents by examining the activations. However, they could be related to rogue dimensions [1], as LLM activation spaces are highly anisotropic.
>
> In conclusion, we thank the reviewer for their constructive comments. Their feedback prompted additional experiments and clarifications that we believe improve both the clarity and robustness of our work. We hope our responses address the reviewer’s concerns.
>
> —
>
> [1] Timkey, William, and Marten Van Schijndel. "All bark and no bite: Rogue dimensions in transformer language models obscure representational quality." (2021).

---

> > ### Comment · Reviewer_59xx · 2025-08-04
> > **Reply**
> >
> > Thank you for your thorough and thoughtful responses.
> >
> > ## W1
> > I appreciate the additional experiment conducted on GPT-2 and acknowledge that the results were replicated across two models. However, I remain hesitant to accept the claim that dense latents are not artifacts of SAE training as generalizable based on only two models. The argument would be significantly strengthened by the inclusion of theoretical analysis supporting the assertion that dense latents are not artifacts. In the absence of such support, I remain unconvinced that this phenomenon is representative or generalizable beyond the current experimental scope.
> >
> > ## Q2
> > Thank you for addressing this question. However, I remain unclear as to why the early stabilization in the number of dense latents necessarily rules out the possibility that they are artifacts of under- or overfitting. Could you elaborate on how this observation excludes the possibility of such training artifacts?
> >
> > ## Others
> > These points are now clear. Thank you for the helpful clarifications.
> >
> > At present, the first part of the paper emphasizes the claim that dense latent representations are not artifacts and tend to exhibit antipodal structure. While this finding is intriguing, I do not find the current evidence sufficient to regard it as a typical or generalizable property. Additionally, concerning the taxonomy, each class of latents appears to be identified using different heuristics, which makes it difficult to form a consistent understanding of intermediate representations in LLMs.
> >
> > For these reasons, I intend to maintain my current score.

---

> > > ### Author Response · Authors · 2025-08-07
> > >
> > > Thank you for reading our response and engaging with the discussion.
> > >
> > > **Regarding W1**: We appreciate the suggestion and will aim to replicate the experiment on LLaMA 3.2 1B for the final version of the paper.
> > >
> > > **Regarding Q2**: The fact that the number of dense latents stabilizes early in training suggests that their emergence is not strongly affected by the amount of training data (i.e., over- or underfitting). If they were a by-product of overfitting, we would expect them to emerge late during training. If they were due to underfitting, we would expect their number to decrease as training progresses. The fact that none of these trends is observed supports the interpretation that dense latents reflect persistent structure in the residual stream rather than training artifacts.
> > >
> > > We thank the reviewer again for their thoughtful feedback.

---

### Official Review · Reviewer_bgXx · 2025-06-29

**Clarity:** 3
**Significance:** 3
**Originality:** 4
**Rating:** 5
**Confidence:** 4

**Summary:**

The paper studies frequently activated latent features (**dense latents**) of SAEs trained on residual stream tokens of language models. The paper defines dense latents as latents active in more than 10% of tokens.

The paper focuses on JumpReLU SAEs trained on Gemma-2-2B layers and includes some experiments with TopK SAEs and GPT-2 small.

The paper finds the following:
- Experiments that ablate the subspace of the residual stream spanned by vectors decoded from dense latents suggest that dense latents are not just results of biases of SAE training.
- For a great proportion of dense latents there is exactly one another latent with encoder cosine similarity close to -1. These are called antipodal pairs. It seems that each pair represents a single direction, and this is not specific to the JumpReLU SAE.
- The role of a substantial proportion of dense latents can be understood to some degree (position tracking, context binding, entropy modulating, letters, part-of-speech-like categories, principal component).

---

**In more detail.** The analysis identifies different roles for 10%-80% of dense latents depending on the LM layer. The roles are categorized as:
1. Position tracking latents: the value depends on the distance from start of sentence, paragraph (in early LM layers) or context (in almost all LM layers).
2. Context-binding latents: approximately consistent meaning within context, but not between contexts. Present in layers in the middle.
3. Nullspace latents: features in the final layers that have a very small effect on the unembedding output, but contribute substantially to the residual stream's norm. Harder to interpret. Some of them modulate entropy by scaling through RMSNorm/LayerNorm [27].
4. Alphabet latents: high-density latents in the last layer that promote tokens that share the initial letter.
5. Meaningful-word latents: latents that are active on tokens that belong to some higher-level part-of-speech based categories.
6. PCA latents: have high cosine similarity with the top principal component of the residual stream.

**Questions:**

**Questions:**
1. Please consider the questions under weaknesses, point 1.
2. Do you know whether the presence of (almost) antipodal pairs is a consequence of the frequency of the latent (more frequent features need to be reconstructed more accurately)? Does reducing the dictionary size reduce the number of antipodal pairs?
3. Do you know how much different types of dense latents overlap?

I apologize for any errors on my part.

**Ethical Concerns:**

["NO or VERY MINOR ethics concerns only"]

**Final Justification:**

The discussion with the authors, which includes clarifications and interesting new empirical information increases my confidence in my positive assessment of the paper.

**Limitations:**

yes

**Paper Formatting Concerns:**

No formatting issues noticed.

**Quality:**

3

**Strengths And Weaknesses:**

**Strengths**

- To my understanding, this paper is a meaningful step in language model interpretability research. It improves the understanding of a substantial proportion of frequently activated SAE latents of language transformer models and gives convincing evidence against the hypothesis that most dense latents are mere artifacts of SAE learning algorithms.
- As far as I can tell, much of the ideas of what to look for and the methodology are original.
- The writing is mostly clear and the experimental results are presented well. (I especially appreciate Figure 2.)
- The source code for reproducing the experiments in the main paper is provided. (I am not sure about the appendix.)

**Weaknesses**

1. Clarity of methodology and presentation could be improved (most important).
	1. Are the inputs of SAEs the features after adding the MLPs output to the residual stream?
	2. Is the subspace ablation based on encoder or decoder vectors?
	3. The definition of "antipodality score" lacks justification (the paper mentions its drawbacks), but I think that it accomplishes its goal.
	4. Relatedly, in Figure 11 I find it unclear how the indices $i, j$ are chosen and whether the antipodality score (color) correspond to $i$ or $j$?
	5. Figure 1. c: Why is encoder cosine similarity chosen rather than decoder cosine similarity?
2. The discussion of related work (background information) could be improved.
	1. Prior work on antipodal pairing [1] and dense latents ("high-frequency latents") [2] should be discussed in relation to the present work.
	2. While the review of recent work seems sufficiently broad (as far as I can tell), it lacks some depth and context. (This is probably due to a lack of space.)
3. Clarity of some claims could be improved.
	1. L36 "antipodal pairs, effectively reconstructing specific subspace directions." Perhaps something like "antipodal pairs, each effectively encoding a single direction" would be more clear.
	2. L45: "inherently dense mechanistic functions" might not be clear before reading section 3.1.
	3. What is the meaning of "a linear combination of the “true” basis works for reconstruction too"?

Minor suggestions:
- Clarity and errors:
	- L78: "many pairs" could be expressed more precisely.
	- Figure 11 a: Consider showing the 10% threshold.
	- Figure 5 b: The boxplots are a bit hard to recognize. What are the dotted lines?
	- L62: typo: ". and" -> ", and"
	- The context of Figure 16 seems to be missing. What are the run lengths of? What is the parameter of the Bernoulli distribution?
	- Consider including an overview of hyperparameters, models and datasets.
- Questions:
	- Would it be interesting to plot distributions of latents (like Figure 11) depending on the frequency?

---

> ### Author Rebuttal · Authors · 2025-07-31
>
> Thank you for acknowledging the originality of the work, and its significance towards understanding of SAEs. Your positive feedback is appreciated. We have addressed each of your questions below.
>
> > **W2:** The discussion of related work (background information) could be improved.
>
> We agree and have significantly expanded the related work section in response. The new version provides a more thorough discussion of the previous studies that are most relevant to our work. The revised section is provided below (with newly added portions highlighted in bold).
>
> —————————————————————————————————————————————
>
> Interpreting SAE Latents.
>
> As SAEs have gained traction, recent work has focused on interpreting the meaning of their latent features [28,44]. Building on the neuron interpretation methodology introduced by [45], several recent works aim to interpret SAE latents systematically. **Templeton et al. [6] propose a rubric-based evaluation method in which a language model (Claude 3 Opus) scores how well a proposed feature description aligns with the contexts on which the latent activates. Similarly, Paulo et al. [46] propose a pipeline in which natural language interpretation for SAE latents are matched with different contexts and used by an LLM in different tasks that evaluate how good the interpretations are in predicting activating and non-activating contexts. Other recent efforts explore automated interpretation approaches based on self-interpretation strategies [47].** A recurring observation across multiple studies is the emergence of a substantial number of dense latents, which activate on more than 10% or even 50% of tokens [8,9]. **Chen and Batson [10] take the 10 most densely activating latents in a cross-layer Transcoder trained on Claude and attempt to manually interpret them, finding plausible interpretations (e.g., "activates on commas", "activates on non-terminal tokens in multi-token words") for 6 of the 10 features. In contrast, Rajamanoharan et al. [11] view dense latents as an undesired phenomenon and propose a frequency-based regularizer to discourage their emergence during training.** Whether these latents reflect meaningful internal computations or arise as undesirable artifacts remains an open question. Our work addresses this question directly, showing that dense latents are persistent, characterized by a specific geometry, and often track interpretable, functional signals in the model.
>
> Dense Language Model Representations.
>
> **Prior work has also identified dense signals in language model representations more broadly (i.e., components that encode information consistently across many tokens). Gurnee et al. [1] present a taxonomy of universal neurons that appear across GPT-2 models trained with different seeds. Among these, they identify neurons that encode positional information at each input token. Chughtai and Lau [3] similarly identify dense positional features in an SAE trained on GPT-2’s layer 0, though they do not explicitly analyze their activation density. Finally, Stolfo et al. [2] describe neurons that regulate model confidence by tracking entropy and connect them to a component of the residual stream aligned with the quasi-nullspace of the unembedding matrix.**
>
> —————————————————————————————————————————————
>
>
> > **W3.3:** What is the meaning of "a linear combination of the "true" basis works for reconstruction too"?
>
> We apologize this was unclear; we had to cut a longer discussion for space reasons. The idea here is that if some set of underlying LLM signals are dense and co-occur perfectly, then any basis of this dense space has the same sparsity penalty to get the same reconstruction (the sparsity penalty $L_0$ is simply the dimension of the dense space), so there is no reason that the "true" basis of features would get learned. Although dense latents are not perfectly co-occurring, some co-occurrence could still cause the learned basis to be slightly rotated from the "true" basis.
>
> > **W5:** Would it be interesting to plot distributions of latents (like Figure 11) depending on the frequency?
>
> Good question. If we color the points in Figure 11c by activation frequency, the trend remains similar: dense latents cluster near (-1, -1), consistent with the antipodality pattern. This follows from the strong correlation between activation frequency and antipodal structure.
>
> > **Q2:** Do you know whether the presence of (almost) antipodal pairs is a consequence of the frequency of the latent (more frequent features need to be reconstructed more accurately)? Does reducing the dictionary size reduce the number of antipodal pairs?
>
> We believe that antipodality is a consequence of the fact that dense features almost always are active, so it is hard for the SAE bias to correctly learn the "bottom" of the dense feature. To support this hypothesis, we see that reducing the size of the dictionary does not significantly reduce the number of pairs (12 pairs at 16k, 18 at 1m), but replacing the ReLU TopK activation function with an "absolute TopK" activation function (allowing activations to be both positive and negative, and taking the top-k activations by absolute value) does remove all pairs.
>
> > **Q3:** Do you know how much different types of dense latents overlap?
>
> Based on our chosen tests and cutoffs, very few (3.6%) of all dense latents fall in more than one category. Most overlaps are between sentence- and paragraph- tracking latents (as discussed in Appendix E.2, Figure 15), as well as between meaningful-word and nullspace/alphabet latents (likely due to the cutoffs chosen).
>
> —————————————————————————————————————————————
>
> We also apologize for the clarity issues. We clarify each one below, and have added the corresponding explanations to the paper.
>
> > **W1.1:** Are the inputs of SAEs the features after adding the MLPs output to the residual stream?
>
> Yes, these SAEs are "between layers", so e.g. the layer 0 SAE is after the first attention and MLP output.
>
> > **W1.2:** Is the subspace ablation based on encoder or decoder vectors?
>
> We ablate based on decoder vectors.
>
> > **W1.3:** The definition of "antipodality score" lacks justification (the paper mentions its drawbacks), but I think that it accomplishes its goal.
>
> Thank you for acknowledging the usefulness of our metric. We designed the antipodality score based on the intuition that antipodal pairs (latents that jointly reconstruct a single line) should exhibit cosine similarity close to -1 in both their encoder and decoder weights. We take the product of these two similarities to capture cases where both alignments are strong and quantify the extent to which a given latent participates in such a pair.
>
> > **W1.4:** Relatedly, in Figure 11 I find it unclear how the indices $i, j$ are chosen and whether the antipodality score (color) correspond to $i$ or $j$?
>
> The indices are indexing SAE latents. For a given SAE latent $i$, we loop over all other SAE latents $j \ne i$, compute the similarity product from equation 1 for the pair of latent i and j, and then pick the latent $j$ that maximizes the quantity in equation 1. This latent $j$ is then used to calculate the antipodality score for latent $i$. Figure 11c then shows this antipodality score for each latent $i$.
>
> >**W1.5:** Why is encoder cosine similarity chosen rather than decoder cosine similarity?
>
> For a single SAE latent, the encoder and decoder directions have high cosine similarity, so it is somewhat arbitrary which we chose to show (i.e., the decoder pattern would be similar).
>
> > **W3.1:** L36 "antipodal pairs, effectively reconstructing specific subspace directions." Perhaps something like "antipodal pairs, each effectively encoding a single direction" would be more clear.
>
> Thank you, we will make this change!
>
> > **W4.1:** "many pairs" could be expressed more precisely.
>
> We will change this to read many pairs ( for the SAE shown in Figure 1, there are 10 such pairs).
>
> > **W4.2:** Figure 11 a: Consider showing the 10% threshold.
>
> To clarify, the plot is showing the median principle angles between each layer’s dense subspace, where the dense subspace is defined as the space spanned by only dense (frequency > 0.1) latents.
>
> > **W4.3:** The boxplots are a bit hard to recognize. What are the dotted lines?
>
> The dots in the boxplots represent outliers outside the whiskers, based on the standard Plotly implementation. Specifically, whiskers extend to 1.5 times the interquartile range, and any data points beyond this range are marked as outliers. We will add a note clarifying this in the appendix of the final version.
>
> > **W4.4:** typo: ". and" -> ", and"
>
> We will fix this, thank you!
>
> > **W4.5:** The context of Figure 16 seems to be missing. What are the run lengths of? What is the parameter of the Bernoulli distribution?
>
> We apologize for this, this is a plot for an additional experiment that we forgot to describe in the text, and we have removed it from the paper (the average run length is the average number of tokens that each latent fires for).
>
> > **W4.6:** Consider including an overview of hyperparameters, models and datasets.
>
> Thank you for this suggestion! We include chosen hyperparameters for feature cutoffs in Appendix E.1 and relevant experiment details in Appendix B, and additionally plan to open source our code.
>
> In conclusion, we thank the reviewer for their thoughtful feedback. We hope that our responses have addressed their concerns and provided sufficient clarification.

---

> ### Comment · Reviewer_bgXx · 2025-08-02
> **Clarification and interesting new insights, increased confidence in positive rating**
>
> Thank you for the response!
>
> I have read all reviews and responses. The authors have clarified many things and provided additional analyses. I find the following particularly interesting:
> - Replacing the ReLU TopK activation of the SAE with an "absolute TopK" activation, which allows negative activations, removes antipodal pairs.
> - Different types of dense latents (according to the chosen thresholds) overlap very little (3.6%).
>
> >> W4.2: Figure 11 a: Consider showing the 10% threshold.
>
> I apologize for the confusion. I was probably thinking of Figure 1, but now I think it is fine either way.

---

> > ### Author Response · Authors · 2025-08-07
> >
> > Thank you for your follow-up and for taking the time to read all the reviews and our responses! We appreciate your engagement.
> >
> > We’re glad you found the additional analyses valuable. We will include them in the final version of the paper.

---

### Official Review · Reviewer_Gno7 · 2025-06-29

**Clarity:** 2
**Significance:** 2
**Originality:** 2
**Rating:** 4
**Confidence:** 3

**Summary:**

The authors study dense latents learned by SAEs trained to reconstruct LLM residual stream representations (with a focus on SAEs pre-trained on Gemma Scope). The authors identify a subset of these latents and classify them to distinct functions. These latents seem to be very scattered in functionality.

**Questions:**

* The antipodality result is interesting, but unfortunately not really connected to the subsequent results of the paper (except a little bit for the context-binding latents). Why is antipodality relevant in the context of density? Why do we see an antipodal pair of features that track 'casino facts' vs 'looking for a buyer'. These concepts do not seem related in any way, except that they both feature in that particular context?

**Ethical Concerns:**

["NO or VERY MINOR ethics concerns only"]

**Final Justification:**

Authors addressed my concerns about the framing of the paper.

**Limitations:**

Yes

**Quality:**

2

**Strengths And Weaknesses:**

Strengths:

* The paper contains some novel analyses and findings regarding the statistics and prevalence of dense SAE features.
* Analyzing dense SAE features is important.
* Classification of dense features is performed, with some novel functionality proposed.
* The idea of dense latents that can bind to relevant context sounds interesting (but unfortunately under-explored).

Weaknesses:
* The paper contains a large list of analyses that are not well connected or seem to bear much relevance to each other. What is the relevance of the antipodality? How are these six different types of dense features related to each other? Some features, like entropy regulating and positional features have been discovered before, as the authors note in the related works section.
* The related works section is extremely brief. The authors mostly just cite work without discussing it.
* For some of the latents (like the concept binding one), it's unclear if this is an actually meaningful latent or an artifact. It's a potentially very interesting functionality - I feel like the authors could have dedicated more space and time to analyze this type of dense latent rather than giving a large list of latents, some of which have been covered before in the literature

In the end, the paper is too unfocused in it's current form. It's not clear if the goal of the paper is to classify as many of these dense features as possible, or to argue that dense features are important/formulate a clear theory about their existence. I think the second strategy is very interesting and I would be willing to raise my score if the authors can expound upon this. The analyses conducted wrt to the concept binding latent is a good start, although it is not clear to me if these bind to distinct types of concepts depending on the context (like latents shown in Fig 3), or if they also specialize on a single concept, but activate densely around it.

One could potentially write a whole paper about dense features like this (if one can indeed show that they have systematic activity patterns, which isn't really that clear from the brief treatment they get in this paper).

---

> ### Author Rebuttal · Authors · 2025-07-31
>
> Thank you for acknowledging the novelty of our analyses and the importance of studying dense SAE features.
>
> > It's not clear if the goal of the paper is to classify as many of these dense features as possible, or to argue that dense features are important/formulate a clear theory about their existence.
>
> Our primary goal is to demonstrate that dense SAE latents are not artifacts of SAEs but instead represent meaningful internal model signals. To support this, we (1) show that these latents do not appear when a particular residual-stream subspace is ablated, and (2) provide functional interpretations for a wide range of them. The classification into six dense-latent types is not the central contribution, but serves as evidence that many dense latents correspond to genuine (and interpretable) dense signals in the language model, rather than SAE artifacts (e.g., aggregated sparse features or optimization noise). This directly challenges the previously hypothesized view of dense latents as optimization byproducts.
>
> That said, we agree with the reviewer that some latent types, such as concept-binding features, deserve deeper analysis. In response, we have conducted new experiments to further explore these, which we describe below.
>
> > [clarifications about context-binding latents] **W3:** “if this is an actually meaningful latent or an artifact.” & “it is not clear to me if these bind to distinct types of concepts depending on the context (like latents shown in Fig 3), or if they also specialize on a single concept, but activate densely around it.” & **Q1:** “Why do we see an antipodal pair of features that track 'casino facts' vs 'looking for a buyer'. These concepts do not seem related in any way, except that they both feature in that particular context?”
>
> Our hypothesis about context-binding latents is that the meaning they encode is context-dependent, rather than a globally monosemantic direction.
>
> Traditionally, SAE features are assumed to activate on a consistent concept (e.g., “Golden Gate bridge”) across all contexts. We found that context-binding latents activate on consistent concepts only *within each context*, but the specific concept they represent *varies across contexts*. For instance, Figure 3 shows #7541 activating on “casino facts” in one document, and “healthcare” in another. This alone does not rule out the possibility that #7541 represents a generic “noun phrase” feature, for example. However, our experiments in Section 4.2 on steering then show that if we amplify such a feature, it influences the generation towards that specific concept, thus it less likely merely represents a generic “noun phrase” feature, but actually contains information about that specific concept.
>
> For an antipodal pair, the two features never co-activate, thus they tend to represent two *distinct* ideas/themes in the passage, for instance “casino facts” and “looking for a buyer”.
>
> To further validate our claim of “context-binding latents activate on specific but varying concepts”, we ran additional auto-interpretability experiments (adding onto Appendix E.5), and will include more examples in the paper, of within-context versus across-context explanations of the latent. Furthermore, if we ask an LLM judge to rate the specificity of explanations, dense latents have within-context explanations that are much more specific than across-context explanations—in contrast with interpretable sparse latents which have similar specificity within-context and across-context.  We will include these additional results in the final version, and thank the reviewer for bringing up the clarification on this claim.
>
> > **W1:** What is the relevance of the antipodality? & **Q1:** Why is antipodality relevant in the context of density?
>
> Antipodality is not the central phenomenon we seek to explain but rather a structural byproduct of how SAEs encode dense residual-stream signals. We hypothesize that antipodal pairs emerge due to the one-sided nature of ReLU activations: a single ReLU unit can capture a feature only if it also learns an appropriate bias value that shifts the activation range to match the full span of the signal. When this signal is dense (i.e., different from 0 most of the time, unlike sparse features) this offset is hard to fit, so the SAE learns a second latent to “complete” the signal, yielding a latent pair whose encoder/decoder weights lie on opposite sides of the same line (i.e., antipodal).
>
> To test this hypothesis, we trained an “Absolute TopK” SAE, where we remove the ReLU to allow the activations of each latent to be both negative and positive. We enforce sparsity by using the TopK most extreme activations for reconstruction. This removes the one-sided constraint and should allow a single latent to encode the entire dense signal. The results confirm our hypothesis: the antipodal structure disappears under this activation. Furthermore, compared to the corresponding TopK SAE, the number of dense latents is smaller and the density of dense latents is higher, as antipodal pairs map roughly to single latents. We will include these results, along with the above explanation, in the final version. We thank the reviewer for prompting this clarification and follow-up experiment.
>
> > **W1:** “Some features, like entropy regulating and positional features have been discovered before, as the authors note in the related works section.”
>
> We would like to clarify that only one of the prior works mentioned in our “Dense Language Model Representations” section actually studies SAE latents: Chughtai and Lau [48] identify positional signals in an SAE trained at GPT-2’s layer 0, but do not analyze their density. The other two works (Gurnee et al. [20] and Stolfo et al. [13]) focus on individual *neurons* in the model and not on SAE features. Stolfo et al., for example, study a confidence-regulation mechanism associated with the unembedding nullspace, but do not relate it to SAEs. We have clarified these distinctions in our revised related work section.
>
> > **W2:** The related works section is extremely brief. The authors mostly just cite work without discussing it.
> We agree and have significantly expanded the related work section in response. The new version provides a more thorough discussion of the previous studies that are most relevant to our work. The revised section is provided below (with newly added portions highlighted in bold).
>
> —————————————————————————————————————————————
>
> Interpreting SAE Latents.
>
> As SAEs have gained traction, recent work has focused on interpreting the meaning of their latent features [28,44]. Building on the neuron interpretation methodology introduced by [45], several recent works aim to interpret SAE latents systematically. **Templeton et al. [6] propose a rubric-based evaluation method in which a language model (Claude 3 Opus) scores how well a proposed feature description aligns with the contexts on which the latent activates. Similarly, Paulo et al. [46] propose a pipeline in which natural language interpretation for SAE latents are matched with different contexts and used by an LLM in different tasks that evaluate how good the interpretations are in predicting activating and non-activating contexts. Other recent efforts explore automated interpretation approaches based on self-interpretation strategies [47].** A recurring observation across multiple studies is the emergence of a substantial number of dense latents, which activate on more than 10% or even 50% of tokens [8,9]. **Chen and Batson [10] take the 10 most densely activating latents in a cross-layer Transcoder trained on Claude and attempt to manually interpret them, finding plausible interpretations (e.g., "activates on commas", "activates on non-terminal tokens in multi-token words") for 6 of the 10 features. In contrast, Rajamanoharan et al. [11] view dense latents as an undesired phenomenon and propose a frequency-based regularizer to discourage their emergence during training.** Whether these latents reflect meaningful internal computations or arise as undesirable artifacts remains an open question. Our work addresses this question directly, showing that dense latents are persistent, characterized by a specific geometry, and often track interpretable, functional signals in the model.
>
> Dense Language Model Representations.
>
> **Prior work has also identified dense signals in language model representations more broadly (i.e., components that encode information consistently across many tokens). Gurnee et al. [1] present a taxonomy of universal neurons that appear across GPT-2 models trained with different seeds. Among these, they identify neurons that encode positional information at each input token. Chughtai and Lau [3] similarly identify dense positional features in an SAE trained on GPT-2’s layer 0, though they do not explicitly analyze their activation density. Finally, Stolfo et al. [2] describe neurons that regulate model confidence by tracking entropy and connect them to a component of the residual stream aligned with the quasi-nullspace of the unembedding matrix.**
>
> —————————————————————————————————————————————
>
> In conclusion, we thank the reviewer for their constructive feedback. Their comments prompted several clarifications (regarding the role of context-binding latents and the broader framing of our contributions), as well as additional experiments on antipodality and context-binding features. We believe these additions substantially strengthen the paper. We hope these improvements will encourage the reviewer to reconsider their evaluation.

---

> > ### Comment · Reviewer_Gno7 · 2025-08-04
> >
> > Thanks for answering my questions and attempting to address my concerns about the framing of the paper. I've increased my score to 4.

---

> > > ### Author Response · Authors · 2025-08-07
> > >
> > > Thank you for reading our response and for raising your score. We truly appreciate your positive feedback.

---

### Official Review · Reviewer_Vwvz · 2025-07-02

**Clarity:** 2
**Significance:** 3
**Originality:** 3
**Rating:** 5
**Confidence:** 3

**Summary:**

This paper investigates the nature of dense latents in sparse autoencoders (SAEs) trained on language model activations. An SAE latent is considered ‘dense’ when firing for at least 0.1 of the inputs. While SAEs are designed to yield sparsely activating, interpretable features, a significant fraction of latents are found to activate on a large number of tokens. The authors challenge the view that these dense latents are undesirable artifacts, instead demonstrating that they often reflect meaningful directions in the residual stream. This paper finds that many of these dense latents capture interpretable information and often fall into one of the following categories: position latents, context-binding latents, (quasi) nullspace latents, alphabet latents, meaningful-word latents, and PCA-latents.  The paper studies how this taxonomy of dense latents changes across model layers. The authors demonstrate that a given latent is in a particular category through a variety of ablation, steering, and quantitative analyses.

**Questions:**

1. How do the types of features recovered by sparse latents compare to those recovered by dense latents? Do they represent different aspects of model computation?


2. What fraction of dense latents do not fall into your taxonomy? Are there properties that reliably predict whether a dense latent is interpretable?

3. In Section 3.1, how exactly is the dense subspace identified and ablated? How large is it, and what is the impact of its ablation on downstream SAE behavior?

4. Could the antipodal structure of dense latents be an emergent by-product of L2 reconstruction loss rather than an interpretable representation structure?

**Ethical Concerns:**

["NO or VERY MINOR ethics concerns only"]

**Final Justification:**

I believe this paper makes a clear contribution and is worthy of publication. I will keep my score of a 5.

**Limitations:**

yes

**Quality:**

3

**Strengths And Weaknesses:**

$\textbf{Strengths}$

- Novel insight into SAE behavior: The paper tackles an underexplored phenomenon—the prevalence and utility of dense latents in SAEs—and provides evidence that these are not just training noise, but reflect intrinsic model structure.

- Careful empirical analysis: The authors use ablations, cosine similarity metrics, and probing techniques (e.g., entropy, steering, AUC prediction from part-of-speech) to characterize dense latents.

- Antipodal structure: The discovery that dense latents form antipodal pairs and correspond to specific directions in the residual stream is a particularly compelling geometric observation.

- Layer-wise functional taxonomy: The paper organizes dense latents into meaningful classes and analyzes their distribution across model layers, revealing a shift from structural to semantic to output-oriented features.

$\textbf{Weaknesses}$

- Lack of geometric contextualization: While the paper makes a strong case that dense latents are not artifacts, it could further contextualize this by considering known geometric properties of LLM representations (e.g., outlier dimensions, early-to-late layer transformations, or anisotropy in the residual stream).

- Sparse vs. dense comparison is missing: The paper focuses heavily on dense latents but does not compare them to sparse latents. Are the features learned by sparse latents of a different nature, or do they partially overlap in function? This comparison would provide crucial context.

- Taxonomy completeness unclear: The paper acknowledges that fewer than half of dense latents fall into its proposed taxonomy. Quantifying this more precisely and identifying predictive signatures of interpretability would strengthen the claims.

- Ambiguous claims: Some empirical observations (e.g., context-binding latents activating on "specific but varying concepts") are not clearly explained and may need clarification or better examples.

---

> ### Author Rebuttal · Authors · 2025-07-31
>
> Thank you for highlighting our “careful empirical analysis” and recognizing the novelty of our study. Your positive feedback is appreciated. We address each of your points below.
>
> > **W1:** While the paper makes a strong case that dense latents are not artifacts, it could further contextualize this by considering known geometric properties of LLM representations
>
> We appreciate this suggestion. We indeed observed a connection between the nullspace-aligned latents and outlier features [1]: the high norm that characterizes outlier features often lies in the unembedding quasi-nullspace, the same subspace tracked by nullspace-aligned latents. We will elaborate on this connection in the final version of the paper.
>
> > **W2:** Are the features learned by sparse latents of a different nature, or do they partially overlap in function? & **Q1:** How do the types of features recovered by sparse latents compare to those recovered by dense latents?
>
> In general, dense latents appear to represent more “functional” or “structural” information (e.g., token position, lexical confidence) rather than semantic content (see Section 5.1 in Chen et al. [2] for a discussion of semantic and structural features). This is consistent with the intuition that semantically interpretable features typically appear sparsely across a text (e.g., a feature representing the concept of “dog” is expected to be active well below 10% of the tokens of a large corpus), whereas functional information related to confidence or token position is expected to be present more densely across tokens.
>
> > **W3:** Taxonomy completeness unclear & **Q2:** What fraction of dense latents do not fall into your taxonomy? Are there properties that reliably predict whether a dense latent is interpretable?
>
> Our classification into six latent types is intended as evidence for the functional roles that dense latents can play. Coverage varies by layer (from 14% to 81% of dense latents per layer) amounting to an overall coverage of roughly 46%. We did not identify simple metrics that predict whether a dense latent is interpretable; interpretation generally requires specific geometric or activation pattern tests, as described in our methodology.
>
> > **W4:** Some empirical observations (e.g., context-binding latents activating on "specific but varying concepts") are not clearly explained and may need clarification or better examples.
>
> Thank you for pointing this out. We will clarify that context-binding latents tend to activate on coherent concepts within the same input, but these concepts vary across inputs. For example, in Figure 3, we see that #7541 activates on “casino facts” in one context, but “healthcare” in another. Furthermore, these context-binding latents come in pairs, and since pairs are antipodal, each tends to activate on a distinct concept from its pair (e.g. #7541 activating on “casino facts”, and its pair #2009 activating on “looking for a buyer”, two concepts that are both present in the context but distinct).
>
> Traditional autointerpretation of SAE latents [3] uses activating examples from the entire corpus (across-context), assuming a specific consistent concept for each latent. We suggest also autointerpreting SAE latents using activating examples within-context. We have since conducted more autointerp experiments (adding on to Appendix E.5), so we will provide more examples in the final version. For many sparse features, if explanations are specific within-context, the explanations are similarly specific with across-context examples (e.g. a feature always for “governmental bodies”). For context-binding latents, the explanations are specific within-context (e.g. “creation, sharing, or replication of digital content”), but vague across-context (e.g. “specific noun phrases, including proper nouns and descriptive phrases”). This is what we mean by “specific but varying”, and we have since quantified this claim further by using an LLM judge to judge specificity. We thank the reviewer for prompting this clarification.
>
> > **Q3:** In Section 3.1, how exactly is the dense subspace identified and ablated? How large is it, and what is the impact of its ablation on downstream SAE behavior?
>
> We identify the dense subspace by collecting the decoder weight vectors of the dense latents from a first SAE. To ablate this subspace from the language model’s residual stream, we remove the component of the residual that lies within it. Formally, let $\mathbf{W} \in \mathbb{R}^{k \times d}$ be the matrix of these weights and $\mathbf{x} \in \mathbb{R}^d$ the residual stream. We compute the projection $\mathbf{p} = \mathbf{x} \mathbf{W}^* \mathbf{W}$ (where $ \mathbf{W}^*$ is the pseudoinverse of $ \mathbf{W}$) and remove it: $\mathbf{x}’ = \mathbf{x} - \mathbf{p}$. This yields the ablated residual stream $\mathbf{x}’$. For comparison, we apply the same ablation procedure using a random subset of sparse latents (ensuring dimensionality matches the dense subspace). In the experiments of Section 3.1, the dimensionality of the ablated dense subspace is 46 in the SAE with 16k latents, and 10 in the SAE with 32k latents. We did not directly quantify the impact of this ablation on downstream model performance, but expect that ablating a persistent, information-carrying subspace would substantially degrade model behavior. We will add an appendix section to clarify this procedure in the final version of the paper.
>
> > **Q4**: Could the antipodal structure of dense latents be an emergent by-product of L2 reconstruction loss rather than an interpretable representation structure?
>
> This is a good observation. We hypothesize that antipodality emerges from the combination of dense signals and the one-sided nature of ReLU activations. Representing a dense signal with a single ReLU unit requires learning a strong negative bias, which the SAE may fail to achieve during training. As a result, the SAE learns a second latent to encode the missing component, forming an antipodal pair.
> To test this, we trained an Absolute TopK SAE by removing the ReLU allowing activations to be both negative and positive, and enforcing sparsity by taking the TopK most extreme activations for reconstruction. This removes the one-sidedness and allows a single latent to capture both sides of a dense signal. As expected, antipodal structure disappears under this activation. We will include this new experiment and explanation in the final version of the paper.
> We thank the reviewer for their detailed and constructive feedback. We hope that our responses have addressed their concerns and provided sufficient clarification.
>
> —
>
> [1] Timkey, William, and Marten Van Schijndel. "All bark and no bite: Rogue dimensions in transformer language models obscure representational quality." (2021).
>
> [2] Chen, Alan, et al. "Transferring Features Across Language Models With Model Stitching." (2025).
>
> [3] Gonçalo Paulo, Alex Mallen, Caden Juang, and Nora Belrose. "Automatically Interpreting Millions of Features in Large Language Models." (2024).

---

> > ### Comment · Reviewer_Vwvz · 2025-08-06
> > **Response to Comment**
> >
> > Thank you for your detailed response and for clarifying some of my questions. After reading the rebuttals and other reviews, I believe my assessment is fair and this paper makes a contribution worthy of publication.

---

> > > ### Author Response · Authors · 2025-08-07
> > >
> > > Thank you for your follow-up and for taking the time to read all reviews and responses. You engagement and positive feedback are greatly appreciated.

---

### Note · Authors · 2025-08-13

We thank the reviewers for the constructive feedback and for the positive overall assessment. Reviewers highlighted the novelty of our analyses (Reviewers Vwvz, Gno7, bgXx) and the evidence that dense latents reflect meaningful residual‐stream structure (59xx, bgXx). They appreciated the antipodal geometry finding (Vwvz, 59xx), and our layer-wise taxonomy showing a shift from structural to output-oriented signals (Gno7, Vwvz).

**What we updated during the discussion**

- We broadened and deepened the related-work section, adding context on LLM geometry (e.g., quasi-nullspace) and SAE interpretation pipelines, and clarified connections to our findings.
- Clarified methodology: SAE placement (between layers); how we identify and ablate the dense subspace (projection via decoder weights and pseudoinverse); the antipodality score (product of encoder/decoder cosine similarities); and figure details (boxplot whiskers/outliers, thresholds).
- Added experiments: replacing ReLU TopK with Absolute TopK removes antipodal pairs and collapses pairs to single latents, supporting the “one-sided activation” hypothesis; replicated dense-subspace ablation on GPT-2 with the same outcome as Gemma 2; added within- vs across-context auto-interpretability results for context-binding latents, showing higher within-context specificity.
- Centralized taxonomy criteria and thresholds (Appendix E.1), explaining our outlier-driven cutoffs with figure references. Coverage varies by layer; overlaps across types are low (~3.6%).

**Additional result addressing Reviewer 59xx’s W1**

To further address Reviewer 59xx’s final comment regarding W1, we replicated the dense-vs sparse-subspace ablation on a third model. In particular, we trained SAEs (d_sae = 8192, k = 32) on LLaMA 3.2 1B’s layer 15. Following the procedure described in Section 3.1, we trained (1) a baseline SAE, (2) an SAE after ablating the dense subspace (density > 0.1), and (3) an SAE after ablating an equally sized sparse subspace. Results follow the same trend observed in Gemma 2 and GPT-2: dense-subspace ablation eliminates dense latents, whereas sparse-subspace ablation leaves the density distribution essentially unchanged (qualitatively matching Figure 1a). This third model strengthens our claim that dense SAE latents are not training artifacts.

---

### Decision · Program_Chairs · 2025-09-17

**Decision:**

Accept (poster)

**Comment:**

This paper studies the dense latent property of the SAE trained on language models.

All the reviewers agree that the analyses are novel and the findings in the paper are original and interesting. The authors successfully address most reviewers' concerns. Although several concerns remain regarding biases arising from the training process and the autoencoder, and the quantitative evidence is not fully sufficient. The reviewers believe the paper is still worth publishing due to its novel results.

I recommend the acceptance of the paper, and encourage the authors to include the discussions and new results in the revision.